



# Variations in river input of iron impact sedimentary phosphorus burial in an oligotrophic Baltic Sea estuary

Wytze K. Lenstra[1], Matthias Egger[1,a], Niels A.G.M. van Helmond[1,2], Emma Kritzberg[3], Daniel J. Conley[2], and Caroline P. Slomp[1]

[1]Department of Earth Sciences –Geochemistry, Utrecht University, PO Box 80021, 3508 TA Utrecht, The Netherlands
[2]Department of Geology, Lund University, SE-223 62, Lund, Sweden
[3]Department of Ecology/Limnology, Lund University, S-223 62, Lund, Sweden
[4]now at: The Ocean Cleanup Foundation, 2624 ES Delft, the Netherlands

**Correspondence:** W.K. Lenstra (w.k.lenstra@uu.nl)

**Abstract.** Estuarine sediments are key sites for removal of phosphorus (P) from rivers and the open sea. Vivianite, an iron (Fe)(II)-P mineral, can act as a major sink for P in Fe-rich coastal sediments. In this study, we investigate the burial of P in the Öre Estuary in the northern Baltic Sea. We find much higher rates of P burial at our five study sites (up to $\sim$0.145 mol m$^{-2}$ yr$^{-1}$) when compared to more southern coastal areas in the Baltic Sea with similar rates of sedimentation. Detailed study of

the sediment P forms at our site with the highest rate of sedimentation reveals a major role for P associated with Fe and the presence of vivianite crystals below the sulfate methane transition zone. By applying a reactive transport model to sediment and porewater profiles for this site, we show that vivianite may account for up to $\sim$40% of total P burial. With the model, we demonstrate that vivianite formation is promoted in sediments with a low bottom water salinity and high rates of sedimentation and Fe oxide input. While high rates of organic matter input are also required, there is an optimum rate above which vivianite

formation declines. Distinct enrichments in sediment Fe and sulfur at depth in the sediment are attributed to short periods of enhanced riverine Fe and organic matter input linked to variations in rainfall on land. Most of the P associated with the Fe in the sediment is likely imported from the adjacent eutrophic Baltic Proper. Our work demonstrates that variations in land-to-sea transfer of Fe may act as a key control on burial of P in coastal sediments. Ongoing climate change is expected to lead to a decrease in bottom water salinity and contribute to continued high inputs of Fe oxides from land, further promoting P burial

as vivianite in the coastal zone of the northern Baltic Sea. This may enhance the role of this oligotrophic area as a sink for P imported from eutrophic parts of the Baltic Sea.

## 1 Introduction

Phosphorus (P) is an important nutrient for primary producers. Burial of reactive P (i.e. bioavailable P) in coastal sediments can permanently remove P from the water column (Froelich et al., 1982; Delaney, 1998; Ruttenberg, 2003). This removal allows

coastal systems to act as a filters for P, reducing the flux of land-derived P to the open sea (Froelich, 1988; Bouwman et al., 2013; Asmala et al., 2017), and as sinks of P imported from the open sea (Asmala et al., 2017). The major sedimentary phases

contributing to the removal of reactive P in coastal sediments are organic-P, iron-bound P and authigenic calcium (Ca)-bound P (Ruttenberg and Berner, 1993; Slomp et al., 1996a).

In most coastal systems, organic matter is a major carrier of P to the sediment (Slomp, 2011). The subsequent microbial degradation of part of the organic matter can lead to elevated concentrations of phosphate ($HPO_4^{2-}$) in the porewater and

precipitation of authigenic P minerals (Froelich et al., 1988; Ruttenberg, 2003). While until recently carbonate fluorapatite was thought to act as the major sink of P in coastal systems (Ruttenberg and Berner, 1993; Slomp et al., 1996b; Ruttenberg, 2003), increasing evidence points towards a significant role for the Fe(II)-phosphate mineral vivianite (Slomp et al., 2013; Egger et al., 2015a; Li et al., 2015; Dijkstra et al., 2016).

Vivianite can form in sediments where the supply of sulfate ($SO_4^{2-}$) is low relative to the input of Fe(oxyhydr)oxides (hence-

forth termed Fe oxides). As a consequence, not all Fe oxides are converted to Fe sulfides (FeS and $FeS_2$) (Ruttenberg, 2003) and Fe oxides can continue to dissolve at depth through Fe reduction coupled to organic matter degradation or anaerobic oxidation of methane ("iron-AOM"; Beal et al. (2009); Egger et al. (2015b); Ettwig et al. (2016)). In coastal sediments, conditions for the formation of vivianite are especially favorable below the sulfate methane transition zone (SMTZ) (Roden and Edmonds, 1997; März et al., 2008; Hsu et al., 2014; Egger et al., 2015a). Here, both porewater $Fe^{2+}$ and $HPO_4^{2-}$ can accumulate in the absence

of sulfide and precipitate as vivianite (Ruttenberg, 2003; Egger et al., 2015a; Dijkstra et al., 2016). Therefore, sediments with high rates of sedimentation and inputs of organic matter and Fe oxides and with an SMTZ that is close to the sediment-water interface are thought to be particularly conducive to vivianite formation (Slomp et al., 2013; Egger et al., 2015a).

Recent studies of diagenesis in sediments of the northern Baltic Sea indicate that variations in the input of organic matter and Fe oxides may contribute to temporal variations in vivianite formation in coastal sediments (Egger et al., 2015a; Rooze

et al., 2016). While Egger et al. (2015a) attributed this to eutrophication, the large magnitude of the change in input of organic matter and of Fe required to describe these data with a reactive transport model (a factor of 15 and 2, respectively; Rooze et al. (2016)) suggests that additional processes are active. We hypothesize that variations in river input of both organic matter and Fe, linked to variations in river discharge, play a role. This hypothesis is based on observed changes in Fe dynamics in soils and rivers in Finland (Sarkkola et al., 2013). More specifically, dry periods may lower groundwater levels exposing Fe

minerals in previously reducing soil layers to oxidation (Laudon et al., 2011). In ensuing wet periods, the freshly formed Fe oxides can be reduced again, after which Fe can be mobilized as $Fe^{2+}$ and transported to rivers and the coastal zone (Sarkkola et al., 2013). While some of the $Fe^{2+}$ will be oxidized to form amorphous and crystalline Fe oxides (Schwertmann and Taylor, 1989; Pokrovsky and Schott, 2002) and will be transported downstream in particulate form, some of the $Fe^{2+}$ can also form complexes with dissolved organic compounds and remain in solution (Rue and Bruland, 1995; Wu and Luther, 1995; Kuma

et al., 1996). Both the particulate and dissolved Fe and the associated organic matter may be transported to the coastal zone through river flow where they may impact P burial.

Future climate change is expected to alter precipitation patterns in Northern Europe and change both the magnitude and seasonality of river discharge (Harley et al., 2006). In the northern Baltic Sea, an increase in the input of freshwater is expected over the coming century (Meier et al., 2006). Such a change may further promote vivianite formation by reducing the

availability of $SO_4^{2-}$ (Jordan et al., 2008; Hartzell et al., 2017). Environmental change may also alter riverine input of Fe. In





the northern Baltic Sea, the input of dissolved Fe from rivers has increased over the past decades (Kritzberg and Ekström, 2012; Sarkkola et al., 2013; Björnerås et al., 2017). The associated increased input of Fe oxides to coastal sediments may also enhance vivianite formation.

In this study, we investigate the burial of P in the Öre Estuary in the northern Baltic Sea, with a specific focus on the factors
contributing to vivianite formation. Porewater and sediment geochemical depth profiles for five sites characterized by varying rates of sedimentation are presented. We apply a reactive transport model to the data for one of the sites and investigate how vivianite formation and P burial in the sediment responds to changes in (1) bottom water salinity, (2) sedimentation rate (3) the input of Fe oxides and (4) the input of organic matter. We also discuss the expected changes in the burial of P in coastal sediments upon ongoing climate change. Our model results suggest that vivianite will become a more important sink for P in
the coastal zone of the Baltic Sea in future because of the expected decline in salinity and the continued high input of Fe from land.

## 2   Methods

### 2.1   Study area and sampling

The Öre Estuary is located along the Swedish coast in the Bothnian Sea, in the northern Baltic Sea (Fig. 1). The estuary is
oligotrophic and has a surface area of approximately 70 $km^2$, a mean depth of 10 m and a bottom water salinity of ca. 5 (www.SMHI.se). The estuary is fed by the Öre River, which has a strongly varying rate of annual discharge (Fig. 2). In this region, the spring flood (April, May) is the major annual hydrological event resulting in a brief period of enhanced input of water, dissolved Fe (Hölemann et al., 2005; Björkvald et al., 2008) and terrestrial organic matter (Rember and Trefry, 2004; Algesten et al., 2006).

This study focuses on sediments from five sites in the Öre Estuary (Fig. 1B). With water depths varying from 10 to 33 m (Table 1). Sediments at all sites are fine-grained and rich in organic matter (Hellemann et al. (2017); Table 1). The major macrofauna species in the area are the mollusk *Limecola balthica* (formerly named *Macoma balthica*), the amphipod *Monoporeia affenis*, the spionid *Marenzelleria* spp. and the isopod *Saduria entomon* (Fig. S.1). *Marenzelleria* first appeared in 1995, but only became abundant after 2002/2003, with densities at our study sites ranging from 187 to 780 $ind\ m^{-2}$ (Kauppi et al.,
2015). The density of *Marenzelleria* spp. in the sediment is water depth dependent with maximum densities at water depths between 30 and 50 meters and lower densities both above and below (Kauppi et al., 2015).

Sediment was collected during two field campaigns with *R/V Lotty* in April and August 2015 using a Gemini gravity corer (8 cm inner diameter). In April, three cores were taken per site. The first core was used for porewater and solid phase analyses, the second core was used for methane ($CH_4$) sampling and the third core for sediment dating using $^{210}Pb$. In August, two cores
were collected per site for porewater analyses and for $CH_4$. At site NB8, a third and a fourth core were collected in August for mineral analyses and additional $^{210}Pb$ measurements.





Methane was sampled directly after core recovery using a core liner with pre-drilled holes with a 2.5 cm depth-spacing. Samples of 10 ml were taken with cutoff syringes from each hole and immediately transferred to a 65 ml glass bottle filled with saturated salt solution. The bottles were stoppered, capped and stored upside down until analysis.

After core retrieval, two bottom water samples were taken and subsequently the cores were sliced into intervals of 1 to 4 cm
under a nitrogen atmosphere at bottom water temperature (Table 1). Each sediment sample was divided over a pre-weighed glass vial for solid phase analysis and determination of water content, and a 50 ml centrifuge tube. The glass vials were stored in nitrogen flushed gas-tight aluminum bags at -20°C until analysis. The 50 ml centrifuge tubes were centrifuged at 3500 rpm for 20 minutes to extract porewater.

## 2.2  Bottom water and porewater analyses

Bottom and porewater samples were filtered through 0.45 μm pore size filters and subsampled under a nitrogen atmosphere. Subsamples were taken for analysis of $SO_4^{2-}$, hydrogen sulfide (where $H_2S$ represents the sum of $H_2S$, $HS^-$ and $S^{2-}$), Fe, manganese (Mn), $HPO_4^{2-}$, ammonium ($NH_4^+$) and dissolved inorganic carbon (DIC). Total Fe and Mn are assumed to represent $Fe^{2+}$ and $Mn^{2+}$, although in the latter case some $Mn^{3+}$ may also be included (Madison et al., 2011). Subsamples for $NH_4^+$ were stored frozen at -20°C. All other subsamples were stored at 4°C until analysis. Samples for $SO_4^{2-}$ were analyzed with ion
chromatography (detection limit of <75 $\mu mol\ L^{-1}$; average analytical uncertainty based on duplicates and triplicates <2%). For $H_2S$, 0.5 ml of porewater was immediately transferred into a 4 ml glass vial containing 2 ml of 2% zinc acetate solution to trap the $H_2S$ as ZnS. Sulfide was determined spectrophotometrically by complexion of the ZnS precipitate in an acidified solution of phenylenediamine and ferric chloride (Cline, 1969). The sulfide standard was verified by titration with thiosulfate. Subsamples taken for porewater $Fe^{2+}$, $Mn^{2+}$ and $HPO_4^{2-}$ were acidified with 10 μL 30% suprapur HCl per ml of sample and
were analyzed by inductively coupled plasma-optical emission spectrometry (ICP-OES; Spectro Arcos).

Concentrations of $NH_4^+$ were determined with the modified indophenol-blue method (Solorzano, 1969). Samples for DIC analyses were collected in 5 mL vials without a headspace and poisoned with 10 μL of saturated $HgCl_2$. Analysis for DIC was performed using an AS-C3 analyzer (Apollo SciTech), consisting of an acidification and purging unit in combination with a LICOR-7000 $CO_2/H_2O$ Gas Analyzer.

Samples for $CH_4$ were prepared for measurement by injecting a 10 ml nitrogen headspace into the bottle. Subsequently, the $CH_4$ concentrations in the headspace were determined by injection of a subsample (50-200 μL) into a Thermo Finnigan Trace GC gas chromatograph (Flame Ionization Detector), after which calculated $CH_4$ concentrations were corrected for sediment porosity. The average analytical uncertainty based on duplicates and triplicates was <3% for DIC and <5% for $CH_4$.

## 2.3  Solid phase analyses

Sediments were freeze-dried and the porosity was determined from the weight loss upon freeze drying. Freeze-dried sediments were ground in an agate mortar under a nitrogen atmosphere and were split into subsamples, that were stored either under oxic or anoxic conditions. For site NB8, the speciation of solid phase Fe, S and P was determined on the anoxic subsamples to avoid oxidation artifacts (Kraal et al., 2009; Kraal and Slomp, 2014). All other analyses were performed on the oxic subsamples.





Ca. 300 mg of sediment was decalcified with two wash steps of 1 M HCl (Van Santvoort et al., 2002) and subsequently dried, powdered and analyzed for carbon using an elemental analyser (Fison Instruments model NA 1500 NCS). Organic carbon ($C_{org}$) contents were determined after correction for the weight loss during decalcification. The average analytical uncertainty based on duplicates and triplicates was <5%. A second subsample of ca. 125 mg was digested in 2.5 ml of $HClO_4$ and $HNO_3$ (ratio

3:2) and 2.5 ml 40% HF in a teflon vessel at 90°C overnight. The acid was evaporated at 140°C until a gel was formed, which was subsequently dissolved in 25 ml of 4.5% $HNO_3$ at 90°C overnight. Total elemental concentrations of Al, Fe, Mn, P and S in the $HNO_3$ solution were determined by ICP-OES, from which total concentrations in the sediment samples were calculated. The average analytical uncertainty based on duplicates and triplicates was <3% for all reported elements.

Sediment samples of ca. 50 mg were subjected to the 4-step Fe speciation procedure of (Poulton and Canfield, 2005) that

targets the following Fe phases: (i) carbonate associated Fe (including siderite and ankerite), extracted for 24 hours with 1 M Na-acetate (pH brought to 4.5 with acetic acid); (ii) easily reducible Fe oxides (including ferrihydrite and lepidocrocite), extracted for 24 hours with 1 M hydroxylamine-HCl in 25% v/v acetic acid; (iii) reducible (crystalline) oxides (including goethite, hematite and akagenéite), extracted for two hours with Na-dithionite (pH 4.8); (iv) recalcitrant Fe oxides (mainly magnetite), extracted for two hours with 0.2 M ammonium oxalate/0.17 M oxalic acid. Samples were measured colorimetrically

using the 1,10-phenanthroline method (APHA, 2005). For simplicity, fractions (ii) and (iii) were summed and henceforth referred to as total Fe oxides. Average analytical uncertainty, based on duplicates, was <5% for all fractions.

Sediment samples of ca. 500 mg were subjected to the 3-step S speciation procedure of (Burton et al., 2006, 2008) that targets the following S phases: (i) acid volatile sulfur (AVS, i.e. mostly FeS) by addition of 2 ml of ascorbic acid and 10 ml of 6 M HCl to the sediment and trapping of the released $H_2S$ into a tube filled with 7 ml Zn-acetate; (ii) elemental sulfur ($S^0$)

by extracting overnight with 25 ml methanol; (iii) chromium reducible sulfur (CRS, i.e. mostly $FeS_2$) by addition of acidic chromium chloride solution and trapping of the released $H_2S$ into a tube filled with 7 ml Zn-acetate. Samples for AVS and CRS were analyzed by iodometric titration of the alkaline Zn-acetate trap. Elemental sulfur was measured colorimetrically according to Bartlett and Skoog (1954). Average analytical uncertainty, based on duplicates, was <3% for all fractions.

Sediment samples of ca. 100 mg were subjected to the SEDEX method described by Ruttenberg (1992) as modified by

Slomp et al. (1996a), but including the exchangeable P step. Five P phases were distinguished: (i) exchangeable-P, extracted for 30 minutes with 1 M $MgCl_2$ (pH 8); (ii) Fe-bound P fraction (including Fe oxide bound P and vivianite; Nembrini et al. (1983); Dijkstra et al. (2014)) extracted for 8 hours with citrate-dithionite-bicarbonate (CDB) (pH 7.5) followed by extraction for 30 minutes with 1 M $MgCl_2$ (pH 8); (iii) authigenic Ca-P (including carbonate fluorapatite, biogenic hydroxyapatite and carbonate-bound P), extracted for 6 hours with Na-acetate buffer (pH 4) followed by extraction for 30 minutes with 1 M $MgCl_2$

(pH 8); (iv) detrital Ca-P, extracted for 24 hours with 1 M HCl; (v) organic P, extracted for 24 hours with 1 M HCl after ashing at 550°C for 2 hours. Steps (i)-(iv) were performed under an argon atmosphere to avoid oxidation artefacts (Kraal et al., 2009; Kraal and Slomp, 2014). CDB solutions were analyzed for P with ICP-OES. All other solutions were measured colorimetrically according to Strickland and Parsons (1972). Average analytical uncertainty, based on duplicates, was <5% for all fractions.



## 2.4 Sedimentation rate and P burial

Sediment accumulation rates at all 5 sites were determined from depth profiles of $^{210}$Pb for April (NB1, N6, N10 and N7) or August (NB8). $^{210}$Pb was measured on freeze dried sediment by direct gamma counting at 46.5 keV using a high purity germanium detector (Ortec GEM-FX8530P4-RB). Self-absorption was measured directly and the detector efficiency was de-

termined by counting a National Institute of Standards and Technology sediment standard. Excess $^{210}$Pb was calculated as the difference between the measured total $^{210}$Pb and the estimate of the supported $^{210}$Pb activity as given by $^{214}$Pb ($^{210}$Pb$_{exc}$ = $^{210}$Pb$_{total}$ − $^{214}$Pb). Sediment accumulation rates at each site were estimated by fitting a reactive transport model (Soetaert and Herman, 2008) to the $^{210}$Pb depth profiles assuming depth dependent changes in porosity and bioturbation (Fig. S.2).

Total P burial (mol m$^{-2}$ yr$^{-1}$) for all sites was calculated as follows:

$$Pburial = P_{total} * sed.rate * \rho * (1 - \phi) * 10^4 \qquad (1)$$

where $P_{total}$ is the averaged concentration of total P (mol g$^{-1}$) over the deepest 10 cm of the sediment sampled, $\phi$ is the porosity in the same interval (vol vol$^{-1}$), sed. rate is the sedimentation rate (cm yr$^{-1}$) and $\rho$ is the density of dry sediment, 2.65 g cm$^{-3}$ (Burdige, 2006).

## 2.5 Scanning electron microscopy (SEM)

Sediment samples from site NB8 were analyzed by SEM to determine whether large vivianite crystals ($\geq$38 μm size) were present. Wet sediments from five depths (sediment depth intervals: 3−4 cm, 14−16 cm, 34−36 cm, 49−52 cm and 55−58 cm) were sieved through a 38 μm mesh size sieve under an argon atmosphere with deoxygenated ultraclean water. The sieved material was washed seven times (5 minutes) with deoxygenated ultraclean water in a sonic bath. After washing, the samples were dried in an argon filled glovebox at ambient temperature. A subsample of the sieved and dried material was mounted on

an aluminum sample holder using double sided carbon tape and subsequently coated with 0.8 mm platinum. Samples were analyzed using SEM-energy dispersive X-ray spectroscopy (EDS; JCM 6000PLUS NeoScope Benchtop SEM) with 15 kV accelerating voltage using a Si/Li detector, in scanning electron mode. Measurements with EDS were performed in the 0−20 keV energy range for elemental quantification (probe current: 1 nA; acquisition time: 50 s (live time)). SEM-EDS software was used to estimate the relative abundances in mol% for the major elements (oxygen (O), sodium (Na), magnesium (Mg),

Al, silica (Si), P, calcium (Ca), Mn, Fe). Samples were detected and photographed in secondary electron imaging (SEI) mode. Measurements were performed with a 1 μm beam in backscatter mode imaging (BEI).





## 3   Reactive transport modeling

### 3.1   General model description

To investigate the mechanisms that control the formation of vivianite in the Öre Estuary a reactive transport model was applied to key porewater and solid phase depth profiles for site NB8. The model describes the mass balance of 11 dissolved and 18

particulate species (Table S.1) and is a modified version of that of Rooze et al. (2016), extended here to include the sedimentary Mn cycle. Three forms of Mn are distinguished, namely Mn oxides, Mn carbonate and $Mn^{2+}$. The transformations of Mn that are included are Mn carbonate and Mn oxide formation and Mn oxide reduction coupled to either $H_2S$, $Fe^{2+}$ or $CH_4$ oxidation (Table S.2; R25-R32).

   The model domain consists of a one-dimensional grid of 900 evenly distributed cells that captures the interval from the

sediment-water interface to a depth of 90 cm. All chemical species are subject to biogeochemical reactions (Table S.2). Solids and solutes are transported by sediment accumulation and bioturbation. Solutes are additionally transported by molecular diffusion and bioirrigation (Soetaert et al., 1996; Wang and Van Cappellen, 1996; Boudreau, 1997). Bioirrigation is modeled as a non-local exchange process (Boudreau, 1984; Emerson et al., 1984).

### 3.2   Model Parameterization

The model was parameterized using data for the field site, information from the literature and by fitting modeled porewater and solid phase depth profiles to the measured data (model constrained; Table S.3). Environmental parameters such as porosity, temperature, salinity and bottom water solute concentrations were measured (Table S.4; Fig. S.3A). The density of dry sediment and the C:N ratio for organic matter were taken from literature (Redfield, 1958; Burdige, 2006). Bioturbation and bioirrigation coefficients were used as fitting parameters (Fig. S.3B&C), taking into account when the most important biorrigator in these

sediments, *Marenzelleria* spp., established (Fig. S.1) and the typical range in these coefficients at similar macrofaunal densities (Renz and Forster, 2013). Fluxes of solids at the sediment-water interface were model constrained. A detailed description of the reactive transport model is given in the supplements.

### 3.3   Iron, manganese and phosphorus dynamics

In the model, Fe oxides and Mn oxides are assumed to consist of fractions with different crystallinities, which affect their

reactivity towards organic matter and $H_2S$ (Table S.1). In both cases, a highly reactive fraction ($\alpha$) and a less reactive fraction ($\beta$) are assumed, whereas for Fe oxides also a refractory fraction ($\gamma$) is included. Only the $\alpha$ fraction of the Fe oxides and Mn oxides is susceptible to reductive dissolution linked to degradation of OM. This allows the $\beta$ fraction to be buried below the zone of organic matter degradation (Rooze et al., 2016).

   Reactive P is deposited at the sediment-water interface in the form of organic-P and Fe oxide bound P. For organic matter

three fractions are assumed: a highly reactive ($\alpha$), less reactive ($\beta$) and refractory ($\gamma$) fraction, following the multi$-$G approach (Jørgensen, 1978; Westrich and Berner, 1984; Middelburg, 1989). The C:P ratio is 300:1 for all three fractions of organic





matter and is model constrained. This relatively high and constant value for the C:P ratio is based on the refractory nature of the organic matter in the area (Stockenberg and Johnstone, 1997; Algesten et al., 2006; Leipe et al., 2011). The three fractions of Fe oxides are assumed to have different P:Fe ratio's, with highly reactive Fe oxides having a higher P:Fe ratio then less reactive crystalline Fe oxides assuming the former can bind more P (Table S.3; Gunnars et al. (2002)). In the bioirrigated

zone, porewater $HPO_4^{2-}$ is assumed to bind to the Fe oxide$^\beta$ fraction present (Fig. S.3C&D) to a maximum P:Fe ratio of 0.28 (mol mol$^{-1}$; Table S.3). This allows for enhanced formation of Fe oxide bound P due to bioirrigation (Norkko et al., 2012). Non-reactive P is deposited at the sediment-water interface as detrital P, authigenic Ca-P, P bound to non-reactive Fe oxides ($\gamma$ fraction) and P bound to non-reactive organic matter ($\gamma$ fraction).

In the sediment, the rate of vivianite formation ($R_{viv}$) is modeled by means of Michaelis-Menten kinetics for dissolved Fe$^{2+}$

and $HPO_4^{2-}$, which implies that the rate of vivianite formation depends on both porewater species and that there is a maximum rate of formation (Reed et al., 2016):

$$R_{viv} = V_{max} \left( \frac{[Fe^{2+}]}{[Fe^{2+}] + K_{Fe^{2+}}} \right) \left( \frac{[HPO_4^{2-}]}{[HPO_4^{2-}] + K_{HPO_4^{2-}}} \right) \tag{2}$$

where $V_{max}$ is the maximum rate (mol L$^{-1}$ s$^{-1}$), $K_{Fe^{2+}}$ and $K_{HPO_4^{2-}}$ are the half saturation constants, [Fe$^{2+}$] and [HPO$_4^{2-}$] are the porewater concentrations of Fe$^{2+}$ and $HPO_4^{2-}$.

## 3.4  Transient modeling scenario

The model was run to steady state for 200 years. Subsequently, temporal changes in the sedimentation rate and the input of organic matter, Fe oxides, Mn oxides and Mn carbonates were implemented to fit key porewater and solid phase depth profiles for site NB8 (Figs. 3 and S.4A-C). The modeled sedimentation rate in the first 200 years was set to 1 cm yr$^{-1}$ (0.34 g cm$^{-2}$ yr$^{-1}$), based on $^{210}$Pb dating (Fig. S.2; Table 1). From 2003 onwards sedimentation was assumed to decrease to 0.6 cm yr$^{-1}$

(0.21 gcm$^{-2}$ yr$^{-1}$; Fig. 3A). The modeled organic matter deposition in the first 200 years was 7.65 mol m$^{-2}$ yr$^{-1}$. From 1987 to 2003, the deposition increased, whereas in the last 12 years the deposition decreased together with the sedimentation rate (Fig. 3B). Our modeled organic matter fluxes are at the high end of the range estimated for the region (1.1 to 8.2 mol m$^{-2}$ yr$^{-1}$; Algesten et al. (2006)). The qualitative trend in organic matter loading assumed in the model is in accordance with results from previous studies on eutrophication in the Bothnian Sea (Fleming-Lehtinen et al., 2008; Fleming-Lehtinen and Laamanen,

2012; Rooze et al., 2016).

The assumption is made that during strongly enhanced riverine Fe oxide input the P:Fe ratio of Fe oxides can be lower because on land the availability of $HPO_4^{2-}$ is not high enough to maintain a high P:Fe ratio. To be able to model a transient P:Fe ratio in the Fe oxide$^\beta$ fraction a fifth Fe oxide fraction, Fe oxide$^{\beta(pulse)}$, was added to the model. This fraction has the same reactivity as Fe oxide$^\beta$ towards organic matter and H$_2$S but has a P:Fe ratio of 0. By varying the ratio between Fe oxide$^\beta$

and Fe oxide$^{\beta(pulse)}$ the P:Fe ratio of the Fe oxide$^\beta$ can be adjusted over time.



In the first 200 years, the Fe oxide input was set to 0.87 mol m$^{-2}$ yr$^{-1}$. The Fe oxide input was then assumed to increase from 1988 onwards until 1997 when a maximum in incoming Fe oxides was assumed, followed by a decrease in Fe oxide input (Figs. 3C). The strong pulse-type increase in Fe oxide input at this maximum was assumed to consist of mainly the $\beta(pulse)$ fraction of Fe oxides (Fig. S.4B). Despite this lower P:Fe ratio, the total flux of P bound to Fe oxides was highest during this

period of enhanced Fe oxide input (Fig. S.4D&E). The transient scenario of Fe oxide loading compares well to qualitative trends used in a previous model study for the same region (Rooze et al., 2016). The initial Mn oxide flux was set to 0.085 mol m$^{-2}$ yr$^{-1}$ (Fig. 3D). In the last 17 years of the scenario, the Mn oxide flux was set to a much lower value. The input flux of Mn carbonate was always low compared to that of Mn oxides (4 - 29% of the total incoming Mn flux; Fig S.4C). Processes coupled to the presence of bioirrigating macrofauna (i.e. bioirrigation and the binding of P onto Fe oxides in the bioirrigated

zone (R32 in Table S.2), were only implemented in the last 12 years of the run, from 2003 onwards.

A model sensitivity analysis was performed to investigate the impact of changes in bottom water salinity, the rate of sedimentation and input of organic matter and Fe oxides on P burial rates and forms. During these sensitivity analyses the transient baseline scenario was used (Fig. 3) and first one (initial) factor was varied per run. Salinity was varied over a range of 0 to 20. The sedimentation rate was varied from 0.25 to 2 cm yr$^{-1}$. The input of organic matter was varied between -70% and +25%;

(3) input of Fe oxides was varied between -25% and +25% from the baseline scenario. Subsequently, a run was performed in which the input of organic matter, Fe oxides and the sedimentation rate were all changed by the same factor as the sedimentation rate to account for the role of rivers as the main source of material in the region (Björkvald et al., 2008; Algesten et al., 2006).

## 4 Results

### 4.1 Porewater profiles

At all sites, depth profiles of porewater constituents showed relatively little difference between April and August 2015 (Fig. 4). Porewater $SO_4^{2-}$ decreased with depth, with a distinct SMTZ only being present in the sampled depth interval at sites N7 and NB8 (Fig. 4). Concentrations of $CH_4$ were low at all other sites. Appreciable $H_2S$ was only present at site NB8 in the SMTZ. Dissolved $Fe^{2+}$ and $Mn^{2+}$ profiles generally showed a maximum near the sediment-water interface and increased concentrations

below ca. 5 to 25 cm depth in the sediment. Porewater profiles of $NH_4^+$ and DIC increased with depth at all sites. Porewater $HPO_4^{2-}$ concentrations were low close to the sediment-water interface and increased with depth at all sites. At sites N7 and NB8, $HPO_4^{2-}$ concentrations remained constant or decreased again below the SMTZ. Concentrations of $Fe^{2+}$, $Mn^{2+}$, $NH_4^+$ and DIC, were highest at depth at site NB8 when compared to the other sites.

### 4.2 Solid phase profiles and sedimentation rate

Organic carbon contents in the surface sediment ranged between 2-4 wt% (Table 1) and decreased with depth at all sites (Fig. 5). Total S concentrations increased with depth in the upper 10 to 20 cm of the sediment and subsequently remained constant



or varied with depth. Total P, total Mn, total Fe and Fe/Al were generally highest close to the sediment-water interface and decreased with depth at all sites except site NB8. Here, several subsurface enrichments in P and Mn were observed directly below maxima in total S, Fe and Fe/Al at depths of 21, 42 and 60 cm.

The Fe and P speciation for site NB8 reveals that the upper 10 cm of the sediment was strongly enriched in Fe oxides and
Fe-bound P. Minima in both Fe oxides and Fe-bound P were found between depths of 10 and 20 cm followed by either constant (Fe oxides) or varying (Fe-bound P) concentrations at greater depth (Figs. 6 and S.5). Maxima in AVS coincided with maxima in total S, total Fe and Fe/Al (Figs. 5&6) and minima in total Mn and Fe-bound P. Concentrations of exchangeable P were highest close to the sediment-water interface but account for only ~2% of total P. Concentrations of detrital P and authigenic Ca-P showed little change with depth.
Sedimentation rates at our study sites varied between 0.225 and 1 cm yr$^{-1}$ (Table 1). Rates of P burial ranged from 0.026 to 0.145 mol m$^{-2}$ yr$^{-1}$ (Table 1). The rates of sedimentation and P burial were highest at site NB8.

### 4.3   SEM-EDS

Examination of sieved sediment fractions for site NB8 with a light microscope revealed the presence of transparent to light blue crystals in sediment intervals below the SMTZ (55−58 cm). Scanning electron micrographs of the sieved fractions revealed
crystals with a similar shape to that of vivianite crystals found in Lake Ørn, Denmark (O'Connell et al., 2015) and the deepest part of the Bothnian Sea (site US5B; Egger et al. (2015a); Fig. 7A) at all sieved intervals below the SMTZ (34−36 cm, 49−52 cm and 55−58 cm). Quantitative analysis of the crystals with electron microprobe-EDS revealed an average Fe:P ratio of 3.3 mol mol$^{-1}$ and the presence of significant amounts of Mn, Mg, Si and Al (Fig. 7B; Table S.5). In samples analyzed above and within the SMTZ such crystals were not observed.

### 4.4   Model results

Modeled porewater and solid phase depth profiles for site NB8 capture the main trends in the measured profiles (Figs. 4, 5 and 6). We note that the model slightly underestimates the amount of Fe-bound P in the upper 15 cm of the sediment and overestimates organic-P. The observed variations in total S, total Mn and Fe-P below 30 cm depth in the sediment were not targeted with the model.
Depth integrated rates of P cycling in 2015 (Fig. 8) show that the release of HPO$_4^{2-}$ from organic-P accounts for only 2.3% of the HPO$_4^{2-}$ release to the porewater at site NB8. The main source of HPO$_4^{2-}$ is the release from Fe oxides, with H$_2$S-driven reductive dissolution accounting for 61% of the HPO$_4^{2-}$ release and dissolution coupled to organic matter degradation accounting for 28% (Fig. 8). In the model, most of the Fe oxide reduction involves the less reactive beta fraction and takes place in the upper 20 cm of the sediment (Fig. S.6; R14&R15).
Approximately 30% of the HPO$_4^{2-}$ removal from the porewater takes place through release to the overlying water by bioirrigation and diffusion, with the former being the most important process. Binding of HPO$_4^{2-}$ to Fe oxides accounts for 60% of the removal while 10% is precipitated as vivianite (Fig. 8). However, Fe oxides that are formed *in-situ* are not a permanent sink





for $HPO_4^{2-}$ as the majority is dissolved through reductive dissolution. Burial of P as vivianite accounts for 41% of the total P burial at site NB8 (Fig. 8).

In the model, total P burial rates increased from 0.12 to 0.34 $\mathrm{mol\ m^{-2}\ yr^{-1}}$ upon a decrease in salinity from 20 to 0 (Fig. 9). Changes in salinity did not affect the burial rates of Ca-P, organic P and P associated with Fe oxides. In the latter case, this was

because only the unreactive fraction of Fe oxides remained. The percentage of P buried as vivianite increased strongly with decreasing salinity.

In the model scenario in which changes in the rates of sedimentation and organic matter and Fe oxide input were coupled, the total P burial rate increased from 0.025 to 0.45 $\mathrm{mol\ m^{-2}\ yr^{-1}}$ when assuming sedimentation rates between 0.25 and 2 $\mathrm{cm}$ $\mathrm{yr^{-1}}$ (Fig. 9). At a sedimentation rate of 0.5 $\mathrm{cm\ yr^{-1}}$, little P was buried in the form of vivianite. At higher sedimentation

rates, formation of vivianite in the sediment increased and the total burial of P was enhanced. Model runs in which changes in the same factors were evaluated separately revealed that increasing the rate of sedimentation or Fe oxide input both enhance vivianite formation and burial of P (Fig. 10A and B; Fig. S.7A&B). The response to an increase in the rate of sedimentation is non-linear, with the greatest increase in P burial between 0.5 and 0.85 $\mathrm{cm\ yr^{-1}}$. For organic matter, the model runs reveal an optimum input rate beyond which P and vivianite burial rates decrease (Fig. 10C; Fig. S.7C).

## 15 5 Discussion

### 5.1 Phosphorus burial in the Öre Estuary

Sediments in the coastal zone of the Bothnian Sea have been suggested to act as an efficient sink for P from rivers and the open sea based on budget calculations and an assumed average burial rate of P of 0.007 $\mathrm{mol\ m^{-2}\ yr^{-1}}$ (Asmala et al., 2017). We find substantially higher rates of P burial in the Öre estuary ranging from 0.026 to 0.145 $\mathrm{mol\ m^{-2}\ yr^{-1}}$ (Table 1). If our data

are representative for the wider area, sediments in the coastal zone of the northern Baltic Sea may be an even more efficient sink for P than previously thought.

In their study, Asmala et al. (2017) identified a linear relationship between P burial and rates of sedimentation for a range of coastal systems around the Baltic Sea, but excluding the northern areas because of a lack of data (Fig. 9). Strikingly, our study sites in the Öre Estuary are characterized by a higher burial of P than predicted by this relationship. This particularly

holds for site NB8. In the following, we will assess whether vivianite formation can play a role in explaining this enhanced P retention. As discussed in detail by Egger et al. (2015a), vivianite formation is generally most pronounced in sediments below the SMTZ, where both $Fe^{2+}$ and $HPO_4^{2-}$ accumulate in the porewater. While we find both solutes in the porewater at depth at all sites, the highest concentrations are observed at sites N7 and NB8 where a distinct SMTZ was present between ca. 10 and 20 cm depth in the sediment. The position of this SMTZ shows little seasonality, likely because of the refractory nature of

the sediment organic matter, which is mostly of terrestrial origin, and the relatively limited seasonal change in bottom water temperature. This implies that any vivianite formed below the SMTZ likely will be preserved and buried, because there will be no further exposure to sulfide.



Detailed analyses of the sediments at site NB8 with SEM-EDS confirm the presence of vivianite crystals below the SMTZ and a lack thereof in and above the SMTZ (Fig. 7). This trend with depth points toward an authigenic origin. The average Fe:P ratio of the crystals of 3.3 mol mol$^{-1}$ (Table S.5) is higher than the 2:1 stoichiometric ratio of vivianite. Similar high ratios were also observed for vivianite crystals in sediments of the Landsort Deep in the Baltic Proper, where they were explained by

surface enrichments of Fe (Dijkstra et al., 2016). The vivianite was also enriched in Mn and Mg, which are both elements that are known to be included in the structure of the mineral (Dijkstra et al., 2016, 2018; Egger et al., 2015a). The presence of Al and Si (Table S.5) likely reflects the presence of clay particles that were not removed prior to the SEM-EDS analysis (Egger et al., 2015a; Dijkstra et al., 2018).

The Fe-P below the SMTZ (Fig. 6) likely is a mixture of both Fe oxide bound P and vivianite, as shown previously for

sediments in the deepest part of the Bothnian Sea (Egger et al., 2015a). The results of the reactive transport model suggest that Fe oxide bound P and vivianite account for ca. 15% and 40% of the total burial of P, respectively. Authigenic Ca-P and organic P concentrations are relatively low and each account for ca. 15% of total P burial (Fig. S.5). The remainder of the P is buried as non-reactive P.

### 5.2 Vivianite formation in coastal sediments in the northern Baltic Sea

Vivianite formation in sediments strongly depends on the balance between the formation of $H_2S$ and the input of Fe oxides (Ruttenberg, 2003). When there is an excess of Fe oxides over $H_2S$, not all Fe oxides will be converted to Fe sulfides, and more $Fe^{2+}$ will be available to precipitate with $HPO_4^{2-}$ as vivianite (Rozan et al., 2002; Gächter and Müller, 2003). The results of the sensitivity analysis (Figs. 9 and 10) show that the bottom water salinity and input of Fe oxides play a critical role in controlling this balance, with vivianite formation and total P burial increasing strongly at lower salinities and with high Fe

oxide inputs. A higher sedimentation rate also enhances the formation of vivianite and P burial because of more rapid burial of Fe oxides below the SMTZ. In our model scenario, sedimentation rates greater than 0.5 cm yr$^{-1}$ were particularly conducive to vivianite formation. Vivianite formation is also highly sensitive to the input of organic matter. When the input of organic matter is lowered to -75% of that assumed in the baseline scenario, the release of $Fe^{2+}$ to the porewater due to Fe oxide reduction decreases to such an extent that formation of vivianite is limited by the availability of $Fe^{2+}$. Consequently, P burial decreases.

An increased input of organic matter relative to the baseline scenario enhances the formation of $H_2S$ and leads to a decline in vivianite formation and P burial. When assuming that changes in the rate of sedimentation and the input of Fe oxides and organic matter are coupled (Fig. 9), the net effect is an increase in vivianite formation and P burial. In summary, conditions for vivianite formation are most favorable in sediments with a low bottom water salinity, a high sedimentation rate and a high input of Fe oxides. While high rates of organic matter input are also required, there is an optimum rate above which vivianite

formation declines.

### 5.3 Variations in riverine Fe and organic matter fluxes

In the Bothnian Sea, boreal rivers are the main source of Fe and organic matter to the coastal zone (Algesten et al., 2006; Björkvald et al., 2008; Palviainen et al., 2015). The input from rivers can be highly variable between years (Rember and Trefry,





2004; Hölemann et al., 2005; Sarkkola et al., 2013). The variations in sediment Fe, S and Fe/Al with depth, with distinct maxima at depths of 21, 42 and 60 cm at site NB8, are in accordance with such a variation in input. This is confirmed by the model, since a strong increase in input of Fe oxides and organic matter is required to describe the maxima in Fe and S in the SMTZ (Fig. 5) and the maximum in vivianite directly below it (Fig. 6). We suggest that these high inputs of Fe and organic

matter are directly linked to variations in river discharge. In 1996, a dry period affected the entire Bothnian Sea region (Fig. S.8; Marttila et al. (2016)). In that year, the Öre River discharge was unusually low (Fig. 2). In the following wet years, i.e. starting in 1997, this then led to an increased input of Fe downstream (Sarkkola et al., 2013). This timing coincides with the maximum in Fe and organic matter input assumed in our model scenario (Fig. 3). In the modeling study for the deep basin site in the Bothnian Sea the timing of the enhanced Fe input also corresponds to the year 1997 (Rooze et al., 2016), implying

that a wide area was affected. Although not included in the model, a similar scenario would explain the enrichments of Fe, S and Fe-bound P at greater depth in the sediment, with the 1976 dry period leading to enhanced input of Fe and organic matter starting in 1977 and the Fe enrichment at 42 cm depth. In summary, our results show that variations in rainfall on land play a critical role in the transport of Fe and organic matter in rivers to the coastal zone. This riverine transport ultimately determines how much P is buried in the coastal zone. In the coastal Bothnian Sea, P burial is twice as high as the input of P from land,

implying that large amounts of the buried P are imported from the open sea (Asmala et al., 2017). Part of the P in the open sea is known to be imported from the adjacent eutrophic Baltic Proper (Savchuk, 2005). The high P burial in the coastal zone coupled to the input of Fe from rivers likely contributes to the seasonal P limitation observed in the Bothnian Sea (Tamminen and Andersen, 2007).

## 5.4   Implications

In most marine systems, the balance between $H_2S$ formation and Fe oxide input is such that most Fe oxides are converted to Fe sulfides, thereby allowing only limited vivianite formation (Ruttenberg, 2003). Here we show, that in estuaries with high inputs of Fe oxides from rivers, high sedimentation rates and sufficient input of organic matter, this balance can become very favorable for the formation of vivianite. Such estuaries can act as a highly effective "coastal filter" for P from rivers and for P imported from the open sea (Bouwman et al., 2013; Asmala et al., 2017).

Our study also highlights the role of bottom water salinity in vivianite formation. Because many studies focus on the relatively high salinity parts of estuaries (e.g. as compiled for the Baltic Sea in Asmala et al. (2017)), the role of vivianite as a sink for P has been largely overlooked.

Sea level rise is expected to increase bottom water salinity in coastal areas in future (Scavia et al., 2002). In the Baltic Sea, however, salinity is expected to decrease because of enhanced precipitation on land and associated higher river runoff (Graham,

2004; Meier et al., 2006). Recent studies also indicate that $Fe^{2+}$ input from rivers in Europe and North America increased over the past few decades, possibly due to changes in land use and expanded forestry (Kritzberg and Ekström, 2012; Björnerås et al., 2017). For the coastal zone of the Baltic Sea, both a decline in salinity and a continued enhanced input of Fe from rivers will promote conditions for vivianite formation in the sediment and increase its role as a sink for P. Detailed studies of other coastal




areas are required to allow a prediction of the role of vivianite formation in the global coastal ocean, since an increased salinity and increased Fe input have opposite effects on the formation of vivianite.

# 6    Conclusions

Our study reveals a very high burial of phosphorus (P) in sediments of an oligotrophic estuary in the northern Bothnian Sea (up to 0.145 mol P m$^{-2}$ yr$^{-1}$). These sediments act as a highly efficient sink for P from rivers and the open sea. We demonstrate that the high P retention in the sediment is related to the formation of vivianite, which is visible in the form of discrete crystals below the sulfate methane transition zone at the site with the highest sedimentation rate. By combining P extractions and reactive transport modeling of solid phase and porewater profiles, we demonstrate that vivianite accounts for 40% of total P burial at the same site. With the model, we also assess the sensitivity of vivianite formation to the key drivers of variations in the availability of H$_2$S versus Fe oxides. We show that vivianite formation is promoted when bottom water salinity is low and rates of sedimentation and inputs of Fe oxides are high. High organic matter inputs are also a requirement, but in our scenario, there is a distinct optimum for vivianite formation.

We suggest that enrichments of Fe, S and P in the sediment are linked to periods of enhanced riverine input of Fe and organic matter. Two sets of enrichments can be coupled to the years 1977 and 1997, when riverine fluxes of Fe were likely enhanced in a wet period directly following a dry period on land in 1976 and 1996, respectively. The enhanced input of Fe and organic matter likely increased the formation of vivianite and the burial of P in the sediment.

In the future, climate change is expected to enhance the freshwater input to the Baltic Sea and thereby decrease its salinity. Continued elevated Fe$^{2+}$ input from rivers is also expected. Both, a decreasing salinity and an increased Fe input will create more favorable conditions for vivianite formation in the sediment. We therefore expect that, in future, vivianite will become more important as a sink for P in the coastal zone of the Baltic Sea.

*Data availability.*   All data will be made available in the database Pangea upon acceptance of the manuscript.

*Code and data availability.*   The model code is available from WL and CS upon request.

*Author contributions.*   CS, WL, NH and DC designed the research. WL, ME, NH and DC performed the analyses. WL performed the model simulations. WL, ME, NH, EK and CS interpreted the data. WL and CS wrote the paper with comments provided by ME, NH, EK and DC.

*Competing interests.*   The authors declare that they have no conflict of interest.





*Acknowledgements.* We thank the captain, crew and scientific participants for their assistance during sampling aboard *R/V Lotty* in April and
August 2015 and D. van de Meent, T. Claessen, T. Zalm, A. van Dijk, D. Kasjaniuk and M. Hermans for analytical assistance. We thank
C. Rabouille for the analysis of porewater $NH_4^+$. We thank M. Hagens for assistance during reactive transport modeling. This research was
funded by the Netherlands Organisation for Scientific Research (NWO-Vici grant 865.13.005; to C.P. Slomp), ERC Starting Grant #278364
5   and this study was partly supported by the BONUS COCOA project (grant agreement 2112932-1), funded jointly by the EU and FORMAS.
The Swedish Meteorological and Hydrological Institute (SMHI) is acknowledged for open source data of macrofaunal abundance and river
discharge.

**Figures**

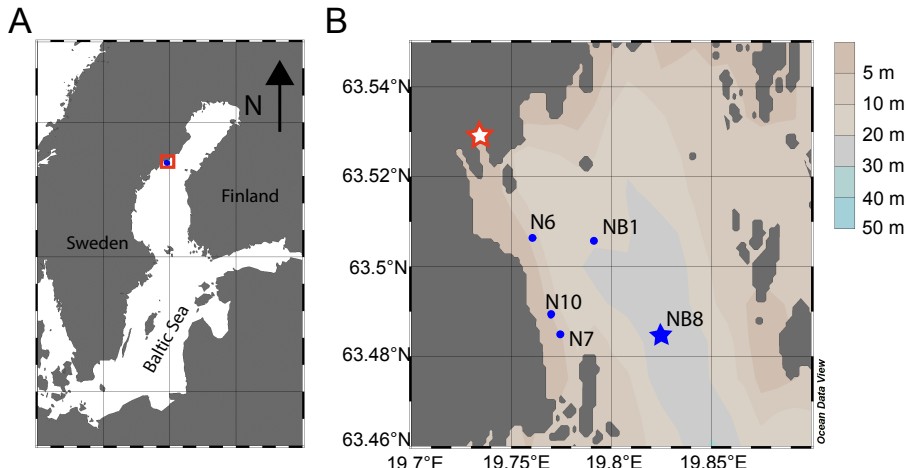

**Fig. 1.** A. The Öre Estuary on the Swedish coast in the Bothnian Sea; B. Locations of the five sampling sites. Site NB8, in the deepest part
of the estuary, is indicated with a blue star. The mouth of the Öre River is indicated with a red star. Figure drawn using Ocean Data View
(Schlitzer, 2015).



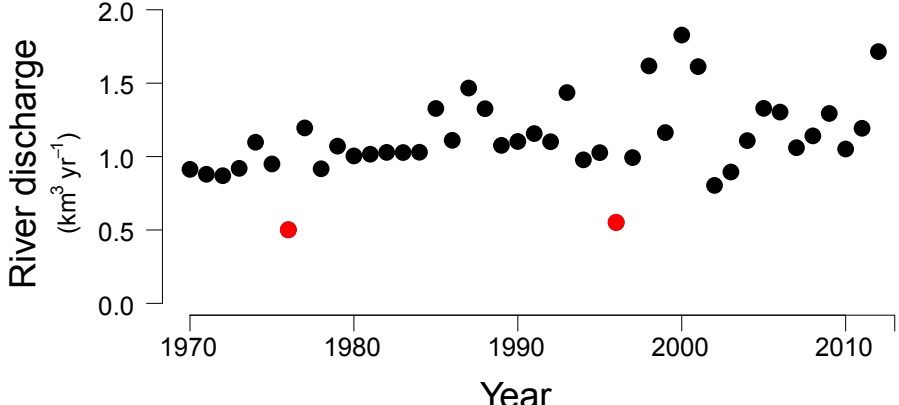

**Fig. 2.** Öre River discharge ($km^3\ yr^{-1}$) from 1970-2012 (Swedish Meteorological and Hydrological Institute (www.SMHI.se)). The red dots indicate two years with an unusually low river run-off, 1976 and 1996, respectively.

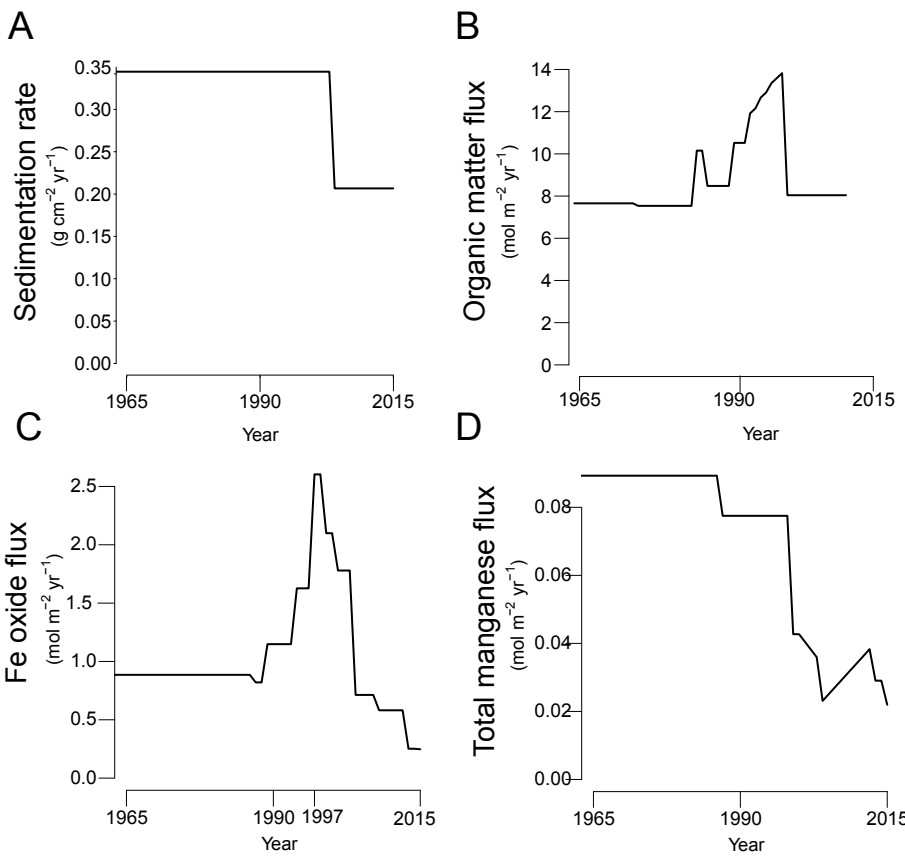

**Fig. 3.** Transient fluxes at the sediment-water interface from 1965-2015 as used in the reactive transport model; A. sedimentation rate; B. organic matter input; C. Fe oxide input. D. total Mn input (Mn oxide and Mn carbonate). Partitioning of different phases of organic matter, Fe oxides and manganese are shown in figure S.4A-C

.





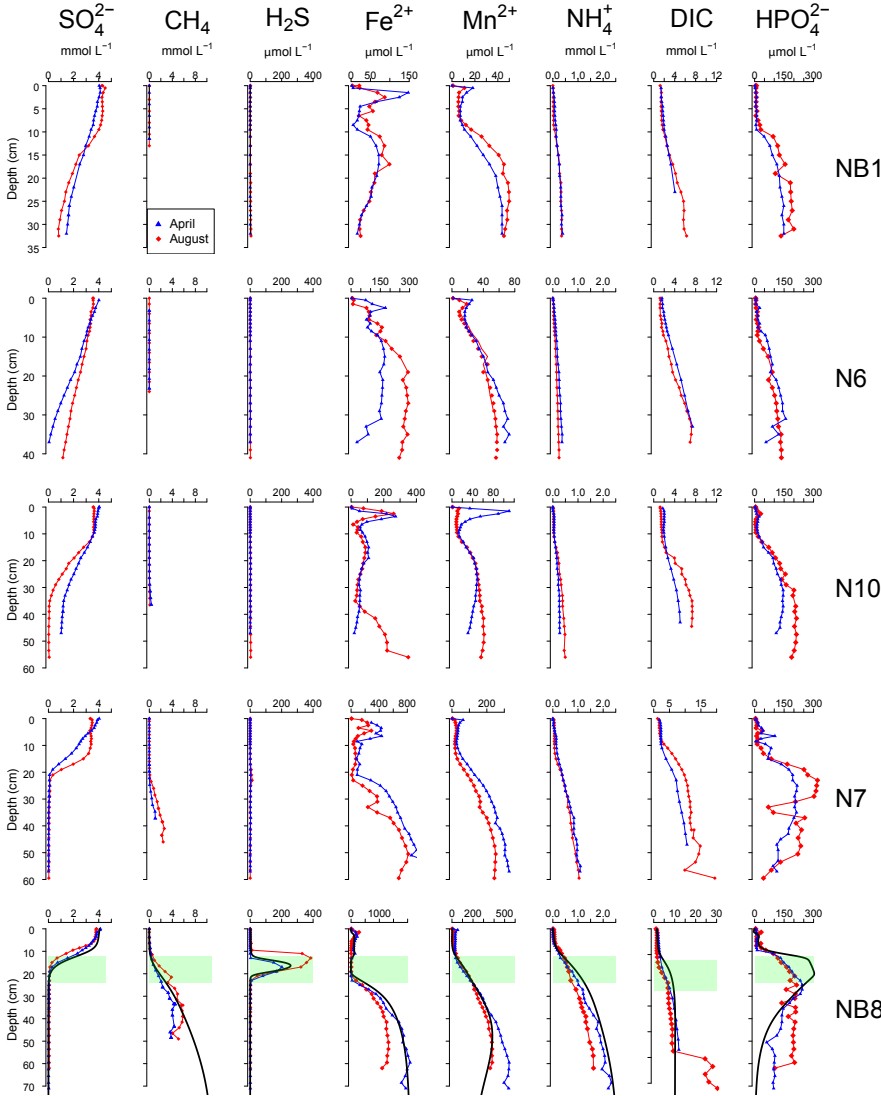

**Fig. 4.** Porewater depth profiles of $SO_4^{2-}$, $CH_4$, $H_2S$, $Fe^{2+}$ and $Mn^{2+}$, $NH_4^+$, DIC and $HPO_4^{2-}$ for sites NB1, N6, N10, N7 and NB8 in April (blue) and August (red) 2015. Porewater profiles of station NB8 include model fits to the April 2015 data for the baseline scenario (black lines). The SMTZ at site NB8 is indicated by the green shaded area.



**Fig. 5.** Solid phase depth profiles of organic carbon, total S, total P, total Mn, total Fe and Fe/Al for sites NB1, N6, N10, N7 and NB8 in April 2015. Solid phase profiles of station NB8 include model fits to all profiles (black lines) except total Fe and Fe/Al for the baseline scenario.



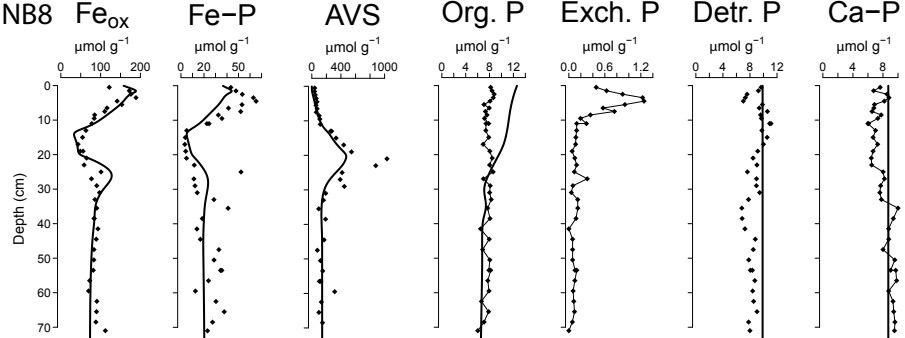

**Fig. 6.** Solid phase depth profiles of Fe oxides, Fe-bound P (Fe-P), AVS, organic-P (Org. P), exchangeable P (Exch. P), detrital Ca-P (Detr. P) and authigenic Ca-P (Ca-P) for site NB8 in April 2015. Solid phase profiles include model fits of the baseline scenario for all profiles except exchangeable P (black lines).

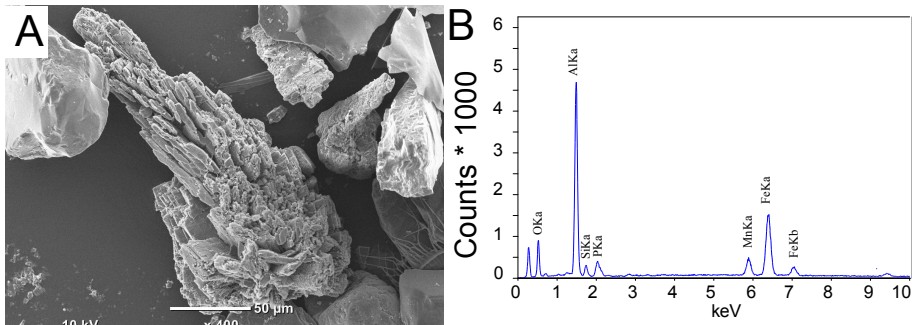

**Fig. 7.** A: Scanning Electron Micrograph (SEM): example of a vivianite crystal from the 34-36 cm depth interval at site NB8 sampled in August 2015. B: Electron microprobe-EDS spectrum of a spot measurement on the surface of the vivianite crystal.





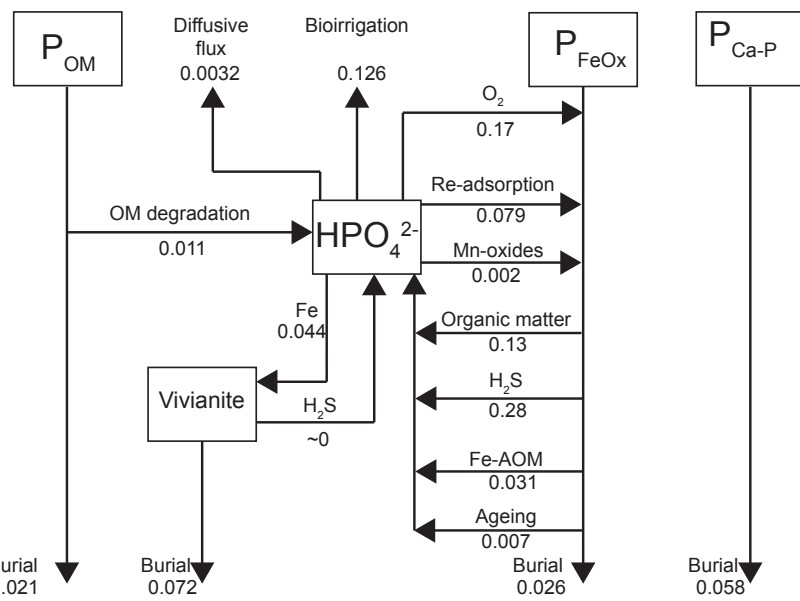

**Fig. 8.** Mass balance of P as calculated with the transient reactive transport model for 2015 (Fig. 3). Results are in non-steady state. Fluxes are in mol P m$^{-2}$ yr$^{-1}$.





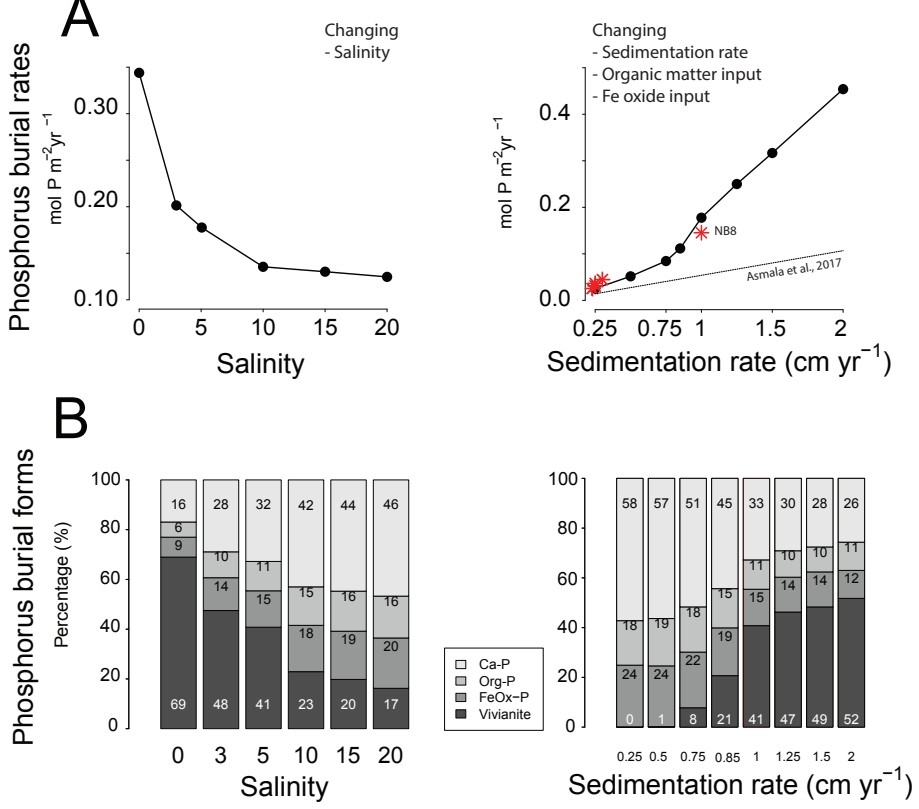

**Fig. 9.** Sensitivity of the rates of A. total phosphorus burial rate and B. the burial forms of P at site NB8 to changes in salinity, as well as in sedimentation rate and inputs of organic matter and Fe oxides as calculated in the model. Four burial forms of P are distinguished: authigenic and detrital apatite (Ca-P), organic-P (Org P), P bound to Fe oxides (FeOx-P) and vivianite. The dashed line indicates the total P burial as a function of sedimentation rate in the coastal zone of the Baltic Sea as suggested in Asmala et al. (2017). Red stars are the burial rates of P as measured at our five sites (Table 1). Sedimentation rates are initial sedimentation rates at t=0.

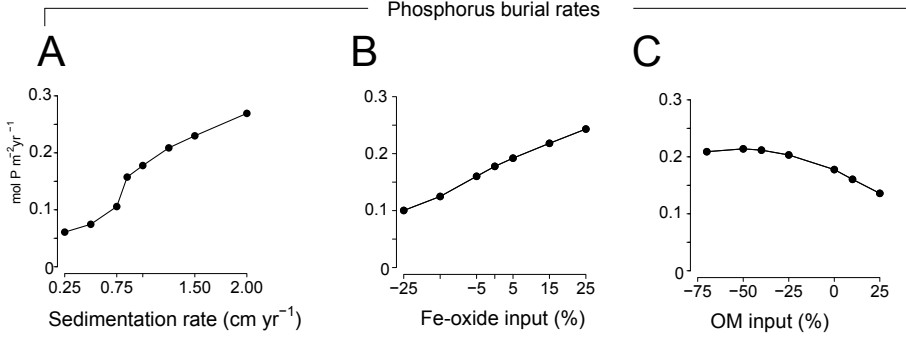

**Fig. 10.** Sensitivity analysis of P burial as a function of A: changing sedimentation rate between 0.25 and 2 cm yr$^{-1}$; B: organic matter input ranging between -70% and +25% C: changing input of Fe oxides ranging between -25% and 25%. The corresponding burial forms of P are given in Figure S.7.





**Table 1.** Water depth, temperature, coordinates and organic carbon at five sites sampled in the Öre Estuary in April and August 2015. Depth unit mbss is meters below sea surface. Bottom water temperatures were derived from CTD data. Total organic carbon ($C_{org}$) is given as the average for the top 2 cm (n=2), with standard deviation between parentheses. Sedimentation rates for sites NB1, N6, N10 and N7 were determined on cores collected in April 2015. The sedimentation rate at site NB8 was determined on a core collected in August 2015.

| Site | Water depth mbss | Temperature °C (Apr/Aug) | Latitude °N | Longitude °E | $C_{org}$ wt. % | Sed. rate cm yr$^{-1}$ | P burial mol m$^{-2}$ yr$^{-1}$ |
|------|------------------|--------------------------|-------------|--------------|-----------------|------------------------|----------------------------------|
| NB1 | 10 | 2.8 / 7.2 | 63.304 | 19.475 | 3.58(±0.14) | 0.225 | 0.026 |
| N6 | 17.2 | 3.2 / 15.2 | 63.303 | 19.454 | 2.06(±0.25) | 0.3 | 0.044 |
| N10 | 20.8 | 2.9 / 7.8 | 63.293 | 19.462 | 3.40±(0.70) | 0.25 | 0.037 |
| N7 | 18.8 | 2.8 / 9.8 | 63.291 | 19.465 | 3.14(±0.30) | 0.25 | 0.031 |
| NB8 | 33.2 | 2.8 / 6.3 | 63.291 | 19.495 | 3.85(±0.07) | 1 | 0.145 |

**Tables Paper**





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
