# Peer review of "Variations in river input of iron impact sedimentary phosphorus burial in an oligotrophic Baltic Sea estuary"

_Biogeosciences, 2018_

## Referee Comment (RC1) · Anonymous Referee #1 · 30 Aug 2018

Overall Quality of the Paper:

This paper makes important contributions to understanding the main drivers of phosphorus burial in coastal sediments, which should be of interest to readers of Biogeosciences. A major finding of this paper is the potential importance of vivianite in the burial of P in estuarine sediments. The paper reports the identification of vivianite crystals in one core sample and presents a model of drivers of P burial that suggests that vivianite may account for about 40% of the P burial at that sample site. The paper also presents a model predicting that vivianite burial increases as salinity decreases and deposition of iron decreases. Therefore, increases in river flow and riverine iron input

clearly have the potential to increase phosphorus burial.

Scientific Questions and Issues:

Title and Abstract:

The title: "Variations in river input of iron impact sedimentary phosphorus burial..." is a bit misleading. I suggest that the authors change the title so that it does not imply that the study directly linked changes in river input of iron to changes in P burial. No measurements of riverine iron inputs are presented. Data on water flow for the Ore River and other Swedish Rivers are presented but it is not entirely clear that profiles of iron and phosphorus in the sediment correlate with periods of high water flow. Two peaks in total Fe concentrations in the sediment are evident at depths where sediment was apparently deposited during years preceded by years of low flow in the Ore River but the flow during the years of deposition was not exceptionally high compared to other years (Fig. 2). The pattern of flow rate for 86 Swedish Rivers combined (Fig. S.8) is somewhat different from that of the Ore River. Finally, a lack of salinity gradient among the sample sites related to their distance from the mouth of the Ore River makes it unclear that discharges of iron from the Ore River would reach the sample sites.

Similarly, lines 10-11 in the abstract are misleading because riverine inputs of iron and organic matter were not measured. Moreover, the claimed increase in river flow is not clear from the data, which mainly show distinct years of low flow preceding the timing of evident enrichments in iron and sulfur. Otherwise, the abstract is clear and reflects the findings of the study.

Section 2.3: Potential for vivianite extraction

This study used a serial extraction method (SEDEX) to characterize five forms of particulate P in sediments (Ruttenburg 1992 modified by Slomp et al. 1996, see p. 5 lines 24-33). One of those extractions uses citrate-dithionite-bicarbonate (CDB) to extract P bound to iron oxides, with the dithionite reducing $Fe^{3+}$ to $Fe^{2+}$. The authors mention

that CDB also extracts vivianite, citing two references for this (p. 5, Line 26). Similarly, Hartzell et al. 2017 (cited on p.2 line 35) suggested that the P extracted with CDB might be mainly from vivianite in samples where $Fe^{2+}$ was abundant and no $Fe^{3+}$ could be detected.

This makes me wonder whether SEDEX could be modified to estimate vivianite concentration separately from iron oxide bound P by preceding the CDB extraction with an extraction using just citrate-bicarbonate. Omitting the dithionite might make this extraction specific to P bound to $Fe^{2+}$, as in vivianite. Having a serial extraction to estimate vivianite concentrations would be very helpful for evaluating the importance of vivianite in P burial. Would the authors comment on this idea?

Section 3.1: Why just model NB8? Model other cores too? How representative is NB8?

On p. 7, Lines 3-4, the authors state that "a reactive transport model was applied to key porewater and solid phase depth profiles for site NB8." Does this mean that the field data to parameterize the model came only from site NB8? This should be clarified on p. 7 line 15. Also it would be good to explain why the model was not applied for other sites as well as NB8. Data points for the other sites plotted on the graph vs. sedimentation rate on Fig. 9A suggest that the model was applied to the other sites, at least partly. Site NB8 is especially interesting because the high rate of sedimentation there suggests a high rate of P burial. How typical are conditions at site NB8? It would be good to address this question somewhere in the paper.

Section 3.4: Why would Fe:P decline with increasing riverine input of Fe?

p. 8 Line 26-27: Why was it assumed that Fe:P would decline if riverine Fe input increased? It seems possible that riverine input of P in eroded soils or suspended riverine sediments might be tightly coupled with riverine input of Fe due to the chemical association of Fe and P in the soils or sediments. Increased river flow might increase particle transport without changing Fe:P. It is interesting to explore the implication of varying the Fe:P ratio either up or down but the authors should explain why increases

in particulate Fe input would not be accompanied by similar increases in particulate P input. One possibility, implied later in the paper, is that the Fe transport is associated more with organic carbon than with suspended soils or sediments.

Sections 5.1 - 5.3: The parallels of FeOx and Fe-P in NB8 core might suggest that the buried Fe-P is bound to FeOx

The profile of FeOx parallels that of Fe-P in core NB8 (Fig. 6) suggesting that Fe-P concentration may be closely tied to the FeOx concentration. Despite this, the model suggests that the Fe-P becomes increasingly associated with $Fe^{2+}$ as depth increases below the SMTZ. This seems a bit puzzling. Does the ratio of FeOx to Fe-P increase with depth? The profiles in Fig. 6 suggest that there might be enough FeOx below the SMTZ to bind the Fe-P. Is the deep FeOx not an effective P sink for some reason? Does FeOx bound P switch to $Fe^{2+}$ bound P (vivianite) below the SMTZ while FeOx persists or increases in concentration? Alternatively, does organic Fe convert to vivianite Fe below the SMTZ?

To understand the change in P partitioning with depth at NB8, I think it would be helpful to see the observed profiles of total Fe (Fig. 5), FeOx and Fe-P (Fig. 6) together in the same figure. Also in the same figure, it would be good to include the profiles of vivianite P and Fe oxide bound P as inferred from the model.

It would also be interesting to illustrate how much of the CDB extracted P (referred to as Fe-P) was associated with $Fe^{3+}$ vs. $Fe^{2+}$ concentrations at site NB8 predicted by the model. This might help assess whether CDB extracts might be useful for quantifying vivianite. Does the sum of FeOx-P and vivianite P equal the Fe-P as measured in the CDB extract? Is there some additional P extracted by CDB or is the sum of FeOx-P and vivianite-P larger than CDB extractable P?

The authors point out maxima in FeOx concentration at 21 cm and 60 cm in core NB8 (Fig. 5). On p. 13 lines 4-7 they attribute those maxima to peaks in Fe input to the estuary happening in 1977 and 1997 following years of low river flow (Fig. 2). They

describe the river flow in 1977 and 1997 as high but actually the flow those years seems close to average, while the preceding years seemed to have distinctly low flow (Fig. 2).

In addition to the concentration maxima at 21 cm and 60 cm, there was a high plateau in the total Fe profile before 1997. This suggests that there may have also been a sustained period of high Fe input before 1997. Is there evidence for sustained high input of Fe before 1997? Alternatively, could the total Fe profile suggest that the sediments may have become less oxidized after 1997, while sediment deposited before then retained a higher portion of refractory FeOx?

Large temporal changes in fluxes to the sediment were inferred from the model and the concentration profiles: Are these realistic?

After running the model to a steady state over 200 years, temporal changes in rates in input of various substances "were implemented to fit key porewater and solid phase depth profiles" (p. 8, line 17). This was also described by Rooze et al. (2016), the source of the model used here. It would be good to add some text describing how the input rates were fitted to the depth profiles. Was it a statistical approach such as regression? If so, are there statistical descriptions of the goodness of fit? The method of fitting was not mentioned in Rooze et al. (2016).

The relative magnitudes of the fitted changes in input rates (Fig. 3 and S.4) were generally larger than the fluctuations in river flows (Figs. 2 and S.8), which are suggested to partly account for the changes in input rates. Sedimentation rate was inferred to drop to a low rate from 2002-2015 (Fig. 3), during a period of relatively high flow for the Ore River (Fig. 2) as well as high flow for other Swedish Rivers (Fig. S.8). It seems more likely that sedimentation rate would increase during high river flow.

In some cases the inferred temporal patterns of inputs for different substances differ inexplicably. For example, input of FeOx peaks sharply from around 1997 when the sedimentation rate drops to a lower level (Fig. 3). Organic matter input follows a similar

pattern. Does this imply that FeOx input is coupled with organic matter input but not with sedimentation rate? The fluctuation in FeOx input seems to drive the assumed change in Fe:P ratios suggesting a partial decoupling of FeOx input from P input. Total Mn input follows a similar pattern as sedimentation rate. Does this suggest that Mn input is more related to sedimentation, while FeOx input is more related to organic matter input?

Inputs of forms of organic matter with different reactivities show opposing temporal changes according to the model fit, with input of refractory organic matter dropping sharply just before 1990 while input of "less reactive" organic matter sharply increases. It may be possible that input of different forms of organic matter with differing reactivities could follow different temporal patterns. However, is it also possible that modeled reaction rates change in an unrealistic way at a certain depth in the sediment?

Adjusting the rates of input of substances to the sediment to fit the concentration profiles is an interesting way to draw inferences about the temporal changes in the input rates. However, it seems possible that errors in modeling reactions at certain depths might lead to erroneous conclusions about temporal variations in inputs to the sediment. The authors should add some discussion of this.

The temporal patterns for inputs of iron and organic matter based on the model in this paper are similar to those in Rooze et al. (2016). However, the model in this paper is adapted from Rooze et al. (2016) so the similarity of the predicted temporal patterns could reflect similar inaccuracies in the model at certain depths in the sediment that could erroneously suggest similar temporal variations in inputs to the sediment.

Instead of adjusting the rates of input of substances to the sediment to fit the concentration profiles the model could assume that the rates of inputs of all substances to the sediment are proportional to the sedimentation rate. I think that the text on p. 9, lines 15-17, indicates that this was done for the sensitivity analysis shown in Fig. 9. If so, it would mean that conclusions based on the analysis in Fig. 9 do not depend on the

validity of the temporal variations in inputs that were fitted to the profiles. I suggest that the authors clarify this.

The effect of salinity:

Section 5.4, p. 13 lines 25-27: This study highlights the role of bottom water salinity. "Because many studies focus on the relatively high salinity parts of estuaries... the role of vivianite as a sink for P has been largely overlooked." This is true. However, Hartzell et al. (2017, cited on p.2 line 35) report SEDEX P fractions along estuarine salinity gradients from 0-11 with sedimentation rates ranging from 0.4-1.8 cm yr-1. Their results were consistent with the model predictions in this paper. Although they did not identify vivianite in their sediments, they proposed that vivianite played an important role in P burial at low salinities in sediments rich in Fe2+.

Detailed comments:

Abstract, lines 8 and 9: Add data on the measured salinity (5) and sedimentation rates (0.25-1.0 cm yr-1) in parentheses.

p. 5 line 6. Clarify what was analyzed by ICP-OES: "...dissolved... ...overnight. [The resulting solution was analyzed for] total element concentrations of..."

p. 9 line 24: "Dissolved Fe2+ and Mn2+ profiles generally showed a maximum near the sediment-water interface." This is in error. Actually, their concentrations were at a minimum near the surface, presumably due to oxidation.

p. 13, line 2: Should cite Fig. 5. Also, maxima at 42 cm are not distinct contrary to the statement in the text.

Fig. S. 5. Profiles of solids: Clarify the caption. Explain what FeOx 1 and FeOx 2 are. Also, note that the caption incorrectly states that the sum of these is graphed.

Consider moving Figs S.5 and S.7 to main text: They seem to present important information (S.5) and important model results (S.7). Fig S.5 seems to belong with Fig. 6.

Fig. S.7. seems to belong with Fig. 10 (which actually cites Fig. S.7. in the caption).

---

## Referee Comment (RC2) · Anonymous Referee #2 · 2 Sep 2018

Review of Lenstra et al. 2018 - Variations in river input of iron impact sedimentary phosphorus burial in an oligotrophic Baltic Sea estuary

Decision I would be happy for this manuscript to be published after some minor corrections.

Manuscript Quality The paper contributes knowledge of phosphorus burial a research largely overlooked in favour of carbon and nitrogen. The research focus on the main drivers of P burial in the northern Baltic sea, though a regional study this work will be of interest to a wide audience and highly relevant to Biogeosciences. The authors report that in this area a significant portion of the P burial is associated with vivianite crystalli-

sation this coupled with they're model outputs highlight an important mechanism for P Burial where increase in fresh riverine water and iron increase P burial.

The hydrological context of this research does need better clarification.

Visual Quality Both the figures and tables are of high quality and are ready for publication.

Technical Quality The methodologies they authors used were appropriate and applied correctly, I cannot comment on modelling. I would have liked to see the core chronologies in the main text not the Sup Mat but Table.1 does provide enough information.

Major Comments Clarify if the flow measurement were made for the Ore river as the title and abstract suggest that the authors are directly linking river input and P burial but this is not supported by the data.

The data from the Ore river does differ from the averaged flows from the 86 other river is there a reason for this. Is the Ore and its catchment an oddity or is it comparable to other estuaries in the area. The authors have cores form 5 sites (NB1,6,7,8 and 10) but only model site NB8. Is there a reason for this and how comparable are the different sites. From Fig.5 it is clear that all the data falls within the same ranges but NB8 is the furthest from the river mouth and a clear statement on why the model was applied to only this site would be useful.

One important question that seems to have not been mentioned is the potential for the Fe-P to be bound to FeOx. I think an additional figure illustrating downcore profile of Total Fe, FeOC, Fe-P and possibly vivianite bound P and organics.

Again line 4 Pg 13 – the refer to the years of 1977 and 1997 as high flow years but both looking at the Ore flow rates and S.Fig8 I would say that they have larger flows. I would focus more of the low flow years proceeding as the major mechanism.

Along the same lines do you have any rainfall data for this period this could be useful in further contextualising the low flows. A quick look at the UEA North Atlantic Oscillation

records (https://crudata.uea.ac.uk/cru/data/nao/nao.dat) both 1976 and 1996 were in the negative phase meaning dry conditions for the higher latitudes. In particular 1996 was in a very strong negative phase (NAO index:-3.27) explaining the low flows. Work completed in Scottish fjords (restricted marine environments not too dissimilar to the research area) showed that during negative NAO phases material builds up in the catchment and when the NAO switches that store of material is quickly transported to the sea – This mechanism may explain the increases in FeOx after the dry/low flow periods.

Gillibrand, PA, Cage, AG & Austin, WEN 2005, 'A preliminary investigation of basin water response to climate forcing in a Scottish fjord: evaluating the influence of the NAO' Continental Shelf Research, vol. 25, pp. 571-587. DOI: 10.1016/j.csr.2004.10.011

The authors do focus on the role of salinity as a key component of the P burial process but as the modelling only takes place at the most saline site is the importance of this overestimated. Clarification would be useful.

Please also note the supplement to this comment:
https://www.biogeosciences-discuss.net/bg-2018-327/bg-2018-327-RC2-supplement.pdf

---

## Author Comment (AC1) · 5 Oct 2018

Overall Quality of the Paper: This paper makes important contributions to understanding the main drivers of phosphorus burial in coastal sediments, which should be of interest to readers of Biogeosciences. A major finding of this paper is the potential importance of vivianite in the burial of P in estuarine sediments. The paper reports the identification of vivianite crystals in one core sample and presents a model of drivers of P burial that suggests that vivianite may account for about 40% of the P burial at that sample site. The paper also presents a model predicting that vivianite burial increases as salinity decreases and deposition of iron decreases. Therefore, increases in river

flow and riverine iron input clearly have the potential to increase phosphorus burial.

Reply: We thank the reviewer for the positive remarks and insightful comments. Below, we provide a point-by-point reply to all the comments and we indicate where and how we revised the manuscript. Scientific Questions and Issues: Title and Abstract:

Comment #1: The title: "Variations in river input of iron impact sedimentary phosphorus burial..." is a bit misleading. I suggest that the authors change the title so that it does not imply that the study directly linked changes in river input of iron to changes in P burial. No measurements of riverine iron inputs are presented.

Reply: We agree with the reviewer that measurements of riverine Fe input would form a useful addition. Unfortunately, long term measurements of total Fe input are not available for the Öre River (or any other rivers in the region) and therefore cannot be included in this manuscript. However, by modelling key porewater and solid phase depth profiles in the Öre Estuary, we demonstrate that large temporal changes in Fe and organic matter input are necessary to fit our model to the measured data at this site (P. 8-9 section 3.4). Because the study area is such an oligotrophic coastal region, the large temporal changes in Fe and organic matter input to sediments located at a river mouth can only be explained by variations in river input. We therefore would strongly prefer to keep our title as it is but refer this point to the editor. If the editor would prefer adaption of the title we would suggest to change it to: "Variations in iron input impact sedimentary phosphorus burial in an oligotrophic Baltic Sea estuary".

Comment #2: Data on water flow for the Ore River and other Swedish Rivers are presented but it is not entirely clear that profiles of iron and phosphorus in the sediment correlate with periods of high water flow. Two peaks in total Fe concentration in the sediment are evident at depths where sediment was apparently deposited during years preceded by years of low flow in the Ore River but the flow during the years of deposition was not exceptionally high compared to other years (Fig. 2).

Reply: We agree that the river flow in the years after each relatively dry period (1976

and 1996) was not exceptionally high. The mechanisms that we propose in this manuscript as a reason for the enhanced input of Fe depend on the relatively low river flow in the years 1976 and 1996 being followed by a higher, more average, flow when directly compared to the preceding dry period. For clarity we changed the text: P2. line 25 from "In ensuing wet periods" to "After dry periods" P.13 line 6 from "In the following wet years," to "in the following years". P14 line 15 from "in a wet period directly following a dry period on land in 1976 and 1996," to "in a period following a dry year on land in 1976 and 1996."

Comment #3: The pattern of flow rate for 86 Swedish Rivers combined (Fig. S.8) is somewhat different from that of the Ore River.

Reply: Spatial and temporal patterns of rainfall over Sweden are expected to differ and result in differences in river discharge between the 86 rivers and the Öre River. Meltwater, for example, plays a more important role for rivers at high latitudes when compared to lower latitudes. As described on P.13 in lines 4-11, the 1996 dry period affected the entire Baltic Sea region. This was less so for 1976.

Comment #4: Finally, a lack of salinity gradient among the sample sites related to their distance from the mouth of the Ore River makes it unclear that discharges of iron from the Ore River would reach the sample sites.

Reply: All sample sites are in close proximity to the river mouth, which is indicated by the star in Figure 1A. There is no salinity gradient in the bottom water, because the Öre Estuary is dominated by a salt wedge. The particle plume from the river varies with time and is known to reach all of our sample sites (Malmgren & Brydsten, 1992).

We now mention this in section 2.1 (P.3 lines 18-19): "While surface water salinity varies with time and distance from the river, there is no salinity gradient in the bottom water of this estuary. The spatial extent of the particle plume from the river varies with time but is known to reach all of our sample sites (Malmgren & Brydsten, 1992). "

Comment #5: Similarly, lines 10-11 in the abstract are misleading because riverine inputs of iron and organic matter were not measured. Moreover, the claimed increase in river flow is not clear from the data, which mainly show distinct years of low flow preceding the timing of evident enrichments in iron and sulfur. Otherwise, the abstract is clear and reflects the findings of the study.

Reply: See reply to comment #1 and #2. We have modified the text (P.1 lines 10-11) to clarify that input of Fe and organic matter is enhanced after dry periods: "Distinct enrichments in sediment Fe and sulfur at depth in the sediment are attributed to short periods of enhanced input of riverine Fe and organic matter. These periods of enhanced input are linked to variations in rainfall on land and follow dry periods."

Comment #6: Section 2.3: Potential for vivianite extraction This study used a serial extraction method (SEDEX) to characterize five forms of particulate P in sediments (Ruttenburg 1992 modified by Slomp et al. 1996, see p. 5 lines 24-33). One of those extractions uses citrate-dithionite-bicarbonate (CDB) to extract P bound to iron oxides, with the dithionite reducing $Fe^{3+}$ to $Fe^{2+}$. The authors mention that CDB also extracts vivianite, citing two references for this (p. 5, Line 26). Similarly, Hartzell et al. 2017 (cited on p.2 line 35) suggested that the P extracted with CDB might be mainly from vivianite in samples where $Fe^{2+}$ was abundant and no $Fe^{3+}$ could be detected. This makes me wonder whether SEDEX could be modified to estimate vivianite concentration separately from iron oxide bound P by preceding the CDB extraction with an extraction using just citrate-bicarbonate. Omitting the dithionite might make this extraction specific to P bound to $Fe^{2+}$, as in vivianite. Having a serial extraction to estimate vivianite concentrations would be very helpful for evaluating the importance of vivianite in P burial. Would the authors comment on this idea?

Reply: We are citing Dijkstra et al. (2014) and Nembrini et al. (1983) here because they demonstrated that vivianite is extracted in the CDB step of the SEDEX method. We agree with the reviewer that an extra step, which would be specific for vivianite, would be helpful to get a better handle on the importance of vivianite in sediments. We

are, however, not aware of a suitable, well-tested step specifically targeting vivianite. We did not test any other extraction methods to only extract vivianite, such as a CDB step without dithionite. Therefore, we cannot comment on this idea.

Comment #7: Section 3.1: Why just model NB8? Model other cores too? How representative is NB8?

Reply: In this study it was our aim to assess the burial of P in and below the SMTZ and the factors contributing to temporal variations in P burial, including the role of the rate of sedimentation. As we were not able to sample below the SMTZ at sites NB1, N6 and N10 (April) and the geochemistry of sites N7 and NB8 is comparable, we focused on site NB8. At site NB8 we subsequently performed three types of chemical extractions and experimental work to allow for visual observations of vivianite. We did not obtain these data for the other sites. The geochemistry at site NB8 is strikingly similar to that at another site with a high sedimentation rate in the Bothnian Sea (Egger et al., 2015a, b), as discussed in the manuscript (e.g. P2, lines 15-17).

We have changed the text in the model description section to clarify that we selected NB8 because it has a shallow SMTZ and a relatively high sedimentation rate (P.7 lines 3-4): "Site NB8 was characterized by a SMTZ close to the sediment-water interface and a relatively high sedimentation rate (Table 1)."

Comment #8: On p. 7, Lines 3-4, the authors state that "a reactive transport model was applied to key porewater and solid phase depth profiles for site NB8." Does this mean that the field data to parameterize the model came only from site NB8? This should be clarified on p. 7 line 15. Reply comment #9:

Reply: We now explicitly mention site NB8 in this sentence (P.7 line 15): "The model was parameterized using data for site NB8,".

Comment #9: Also it would be good to explain why the model was not applied for other sites as well as NB8.

Reply: Please see our reply to comment #7.

Comment #10: Data points for the other sites plotted on the graph vs. sedimentation rate on Fig. 9A suggest that the model was applied to the other sites, at least partly. Site NB8 is especially interesting because the high rate of sedimentation there suggests a high rate of P burial. How typical are conditions at site NB8? It would be good to address this question somewhere in the paper.

Reply: We indicated in the caption of figure 9 that these are measurements, i.e. red stars are used to indicate the burial rates of P as measured at our five sites (Table 1). We now also include this information in the figure itself. Regarding the representativeness of NB8: we refer to our reply to comment #7.

Comment #11: Section 3.4: Why would Fe:P decline with increasing riverine input of Fe? p. 8 Line 26-27: Why was it assumed that Fe:P would decline if riverine Fe input increased? It seems possible that riverine input of P in eroded soils or suspended riverine sediments might be tightly coupled with riverine input of Fe due to the chemical association of Fe and P in the soils or sediments. Increased river flow might increase particle transport without changing Fe:P. It is interesting to explore the implication of varying the Fe:P ratio either up or down but the authors should explain why increases in particulate Fe input would not be accompanied by similar increases in particulate P input. One possibility, implied later in the paper, is that the Fe transport is associated more with organic carbon than with suspended soils or sediments.

Reply: We agree with the reviewer that enhanced input of Fe could lead to enhanced input of P in agricultural environments where reactive P is abundantly present. However, as the area around the northern Baltic Sea consists mainly of natural and not agricultural land, changes in the P:Fe ratio are expected to occur when concentrations of Fe strongly increase. Moreover, we were not able to model the key porewater and solid phase depth profiles when assuming an invariant P:Fe ratio.

We have now added text on page 8 to explain this (line 27): "This assumption is based

on the dominance of natural forest and wetlands in the region (Björkvald et al., 2008). We note that the model results were very sensitive to the P:Fe ratio and that we could not model the key profiles when assuming a higher P:Fe ratio".

Comment #12: Sections 5.1 - 5.3: The parallels of FeOx and Fe-P in NB8 core might suggest that the buried Fe-P is bound to FeOx. The profile of FeOx parallels that of Fe-P in core NB8 (Fig. 6) suggesting that Fe-P concentration may be closely tied to the FeOx concentration. Despite this, the model suggests that the Fe-P becomes increasingly associated with Fe2+ as depth increases below the SMTZ. This seems a bit puzzling. Does the ratio of FeOx to Fe-P increase with depth? The profiles in Fig. 6 suggest that there might be enough FeOx below the SMTZ to bind the Fe-P. Is the deep FeOx not an effective P sink for some reason? Does FeOx bound P switch to Fe2+ bound P (vivianite) below the SMTZ while FeOx persists or increases in concentration?

Reply: We agree with the reviewer that there are strong parallels between the FeOx and Fe-P profile at site NB8. However, these parallels are mainly found in and above the SMTZ where vivianite plays a minor role. Below the SMTZ, FeOx slightly declines with depth, whereas Fe-P concentrations are variable. We realize that our terminology could be misunderstood here given that the extraction steps for Fe-oxides (Feox) could also extract Fe from vivianite. We have now changed Figure 6 and its caption and our method section to convey that vivianite Fe is also extracted in the HCl and CDB steps (P.5, lines 11-13).

There is indeed a sink switch between Fe oxide bound P and Fe(II)-P minerals below the SMTZ. This is explained in the introduction (P2, lines 12-17) and is also discussed in detail by Egger et al. (2015a). For clarity, we have now added text in section 5.1 briefly explaining the source of the Fe2+ and HPO42- in the porewater (P.11 lines 30-32): "As discussed in detail by Egger et al. (2015a), vivianite formation is generally most pronounced in sediments below the SMTZ, where both Fe2+ and HPO42– accumulate in the porewater due to dissolution of Fe-oxides and release of associated P".

We have now also added a figure to the supplements (Fig. S.7) showing this relative distribution, both for Fe and P, as calculated with the model and corresponding text in the results (P.10, line 29): "At depth, P bound to vivianite becomes more important compared to P bound to Fe oxides (Fig. S.7)." The relative contribution of Fe-oxide bound P and vivianite is discussed in lines 9-11 on page 12: "Due to dissolution of Fe oxides and vivianite formation in the sediment the fraction of P bound to vivianite increases with depth (Fig. S.7)."

Comment #13: Alternatively, does organic Fe convert to vivianite Fe below the SMTZ?

Reply: All Fe species that can be mobilized to the pore water potentially contribute to the formation of vivianite. In the applied Fe extraction we did not include a step that was specific for organic Fe. However, labile organic Fe complexes are expected to be extracted in both Na-acetate and hydroxyl-HCl (step 1 and step 2; Jilbert et al., 2018). This labile organic Fe fraction is expected to be small.

Comment #14: To understand the change in P partitioning with depth at NB8, I think it would be helpful to see the observed profiles of total Fe (Fig. 5), FeOx and Fe-P (Fig. 6) together in the same figure. Also in the same figure, it would be good to include the profiles of vivianite P and Fe oxide bound P as inferred from the model. It would also be interesting to illustrate how much of the CDB extracted P (referred to as Fe-P) was associated with Fe3+ vs. Fe2+ concentrations at site NB8 predicted by the model. This might help assess whether CDB extracts might be useful for quantifying vivianite. Does the sum of FeOx-P and vivianite P equal the Fe-P as measured in the CDB extract? Is there some additional P extracted by CDB or is the sum of FeOx-P and vivianite-P larger than CDB extractable P?

Reply: Since the HCl and CDB-extractable Fe is only a relatively small proportion of the total Fe and includes Fe in both vivianite and Fe-oxides (see reply to comment #12), we do not see what can be gained from repeating the total Fe profile in Fig. 6. We do agree that it would be very helpful to include a figure with the partitioning of Fe-bound

P and Fe from the model and have done so in reply to comment #12, i.e. we have added this information as a figure in the supplements (Fig. S.7).

Comment #15: The authors point out maxima in FeOx concentration at 21 cm and 60 cm in core NB8 (Fig. 5). On p. 13 lines 4-7 they attribute those maxima to peaks in Fe input to the estuary happening in 1977 and 1997 following years of low river flow (Fig. 2). They describe the river flow in 1977 and 1997 as high but actually the flow those years seems close to average, while the preceding years seemed to have distinctly low flow (Fig. 2).

Reply: This has been changed. Please see our reply to comment #2.

Comment #16: In addition to the concentration maxima at 21 cm and 60 cm, there was a high plateau in the total Fe profile before 1997. This suggests that there may have also been a sustained period of high Fe input before 1997. Is there evidence for sustained high input of Fe before 1997? Alternatively, could the total Fe profile suggest that the sediments may have become less oxidized after 1997, while sediment deposited before then retained a higher portion of refractory FeOx?

Reply: We have no evidence for a sustained period of high input of Fe prior to 1997. Instead, increased diagenetic remobilization of Fe following the enhanced organic matter input likely contributes to the contrast in total Fe above and below 21 cm. Indeed, sediments deposited prior to 1997 will have retained a higher portion of Fe oxides. We now mention in the discussion that enhanced input of Fe and organic matter enhances conversion of Fe oxides to FeS (P.13 lines7-8): "The increased input of Fe oxides and organic matter leads to enhanced conversion of Fe oxides to Fe sulfides."

Comment #17 Large temporal changes in fluxes to the sediment were inferred from the model and the concentration profiles: Are these realistic?

Reply: Yes, we believe these are realistic, given that stream fluxes of total Fe and Mn in northern Sweden are known to be highly variable and show even larger seasonal

changes (Björkvald et al., 2008). We added this information in the text (P.3 lines 19-23).

In this region, the spring flood (April, May) is the major annual hydrological event. The spring flood results in a brief period of enhanced water flow in streams and rivers (Hölemann et al., 2005; Björkvald et al., 2008) and export of Fe and terrestrial organic matter to the coastal zone (Rember and Trefry, 2004; Algesten et al., 2006). During high flow, total and dissolved Fe and Mn in streams can increase by a factor of up to 10, with concentrations being highly variable between years (Björkvald et al., 2008).

Comment #18: After running the model to a steady state over 200 years, temporal changes in rates in input of various substances "were implemented to fit key porewater and solid phase depth profiles" (p. 8, line 17). This was also described by Rooze et al. (2016), the source of the model used here. It would be good to add some text describing how the input rates were fitted to the depth profiles. Was it a statistical approach such as regression? If so, are there statistical descriptions of the goodness of fit? The method of fitting was not mentioned in Rooze et al. (2016).

Reply: We did not use a statistical approach to fit the input rates to the depth profiles. The model was fitted to the data by visual observation. This is an approach that is applied broadly; e.g.: - Berg et, al. 2003; Dynamic Modeling of Early Diagenesis and Nutrient Cycling . A Case Study in an Arctic Marine Sediment; American Journal of Science - Reed et, al. 2011; Sedimentary phosphorus dynamics and the evolution of bottom-water hypoxia: A coupled benthic-pelagic model of a coastal system; Limnology and Oceanography - Dale et, al. 2013; Modeling benthic–pelagic nutrient exchange processes and porewater distributions in a seasonally hypoxic sediment: evidence for massive phosphate release by Beggiatoa? - Reed et, al. 2011; A quantitative reconstruction of organic matter and nutrient diagenesis in Mediterranean Sea sediments over the Holocene; Geochimica et Cosmochimica Acta We now added this information on P.7 lines 15-16: "The model was parameterized using data for NB8, information from the literature and by visually fitting modeled porewater and solid phase

depth profiles to the measured data (model constrained; Table S.3)."

Comment #19: The relative magnitudes of the fitted changes in input rates (Fig. 3 and S.4) were generally larger than the fluctuations in river flows (Figs. 2 and S.8), which are suggested to partly account for the changes in input rates. Sedimentation rate was inferred to drop to a low rate from 2002-2015 (Fig. 3), during a period of relatively high flow for the Ore River (Fig. 2) as well as high flow for other Swedish Rivers (Fig. S.8). It seems more likely that sedimentation rate would increase during high river flow.

Reply: Please see our reply to comment #2. Chemical processes on land are suggested to be the main driver of these changes (described on P2. lines 22-31). As a consequence, changes in input rates of elements can be larger than fluctuations in the river flow and may be partly decoupled from rates of sedimentation. The period of 2002-2015 is not characterized by an exceptionally high river flow. After the dry year in 1996, the flow returned to an average river flow, therefore no large changes in sedimentary input are expected.

Comment #20: In some cases the inferred temporal patterns of inputs for different substances differ inexplicably. For example, input of FeOx peaks sharply from around 1997 when the sedimentation rate drops to a lower level (Fig. 3). Organic matter input follows a similar pattern. Does this imply that FeOx input is coupled with organic matter input but not with sedimentation rate?

Reply: Please see our reply to comment #19. Indeed, Fe oxides and organic matter are strongly interlinked, as described on P.2 lines 27-31. In this oligotrophic estuary, where the Öre River is the main source of both Fe and organic matter (P.12 lines 32-33), a similar pattern between input of Fe oxides and organic matter is expected. Because chemical processes on land are thought to be responsible for the variations in both the Fe and organic matter flux, this is not necessarily coupled to the input of other sedimentary material.

Comment #21: The fluctuation in FeOx input seems to drive the assumed change in

Fe:P ratios suggesting a partial decoupling of FeOx input from P input. Total Mn input follows a similar pattern as sedimentation rate. Does this suggest that Mn input is more related to sedimentation, while FeOx input is more related to organic matter input?

Reply: Processes on land can affect river fluxes of Fe to the coastal zone as described in the introduction (P.2 lines 22-31). However, this mechanism is probably different for manganese (Mn). While reduced species of Fe are expected to be oxidized during a dry period, in the case of Mn less reduced species are expected to be formed resulting in limited re-oxidation of Mn during a dry period. We have added a sentence to explain this difference in riverine Fe and Mn input after a dry period: (P.13 lines 12-14): "In contrast to Fe, limited re-oxidation of Mn is expected during a dry period because of less formation of reduced Mn species (Burdige 1993). Therefore, an increase in riverine Mn input is not expected after a dry period." The coupling between Fe oxides and organic matter is described in the reply to comment #20.

Comment #22 Inputs of forms of organic matter with different reactivities show opposing temporal changes according to the model fit, with input of refractory organic matter dropping sharply just before 1990 while input of "less reactive" organic matter sharply increases. It may be possible that input of different forms of organic matter with differing reactivities could follow different temporal patterns. However, is it also possible that modeled reaction rates change in an unrealistic way at a certain depth in the sediment?

Reply: The degradation kinetics of the three types of organic matter are restricted by the need to be able to fit porewater profiles of ammonium, dissolved inorganic carbon and methane, which are all determined by the degradation of organic matter. Further constraints are provided by the reaction kinetics and parameters from the literature (as described in Table S.7). This greatly limits the degrees of freedom while fitting. This is explained in detail by Van Cappellen and Wang (1996). We now added a sentence to the supplements to explain this (P.1 line 35): "The large number of model components, the parameters specific for the field site and the reaction kinetics and parameters from the literature greatly limit the degrees of freedom while fitting the model to the data

(Van Cappellen and Wang 1996)."

Comment #23: Adjusting the rates of input of substances to the sediment to fit the concentration profiles is an interesting way to draw inferences about the temporal changes in the input rates. However, it seems possible that errors in modeling reactions at certain depths might lead to erroneous conclusions about temporal variations in inputs to the sediment. The authors should add some discussion of this.

Reply: Please see our reply to comment #22 and reference to the work of Van Cappellen and Wang (1996) in the supplements (P.1 line 35). The kinetics of most chemical reactions used in the model are constrained by the literature (Table S.7). Further constraints are provided by the specific parameters for the field site and the large number of model components and reactions that are modelled simultaneously and that are known to occur in these types of sediments. This makes the results robust. The approach we are using here is not unusual and has been applied in other studies to infer temporal changes in boundary conditions at the sediment-water interface (e.g. Berg et al., 2003; Reed et al., 2011; GCA, Dale et al., 2013). We find it difficult to identify and include consequences of "possible errors in modeling reactions" since there are no obvious candidates for such reactions.

Comment #24: The temporal patterns for inputs of iron and organic matter based on the model in this paper are similar to those in Rooze et al. (2016). However, the model in this paper is adapted from Rooze et al. (2016) so the similarity of the predicted temporal patterns could reflect similar inaccuracies in the model at certain depths in the sediment that could erroneously suggest similar temporal variations in inputs to the sediment.

Reply: The specific model-code used in this study was adapted from Rooze et al. (2016). However, this is a standard reactive transport model using sets of reactions and transport equations that were first combined in such multi-component models more than two decades ago (e.g. Wang and Van Cappellen 1996 and additional references in

the appendix) and have since then been used in a multitude of similar models. Reaction kinetics are based on extensive literature (Table S.7). This means that our results are not specific to this model code. We now have added a sentence in the model description section to explain this (P.7 line 6): "This is a standard multi-component reactive transport model based on principles outlined by, for example, Wang and Van Cappellen (1996)."

Comment #25: Instead of adjusting the rates of input of substances to the sediment to fit the concentration profiles the model could assume that the rates of inputs of all substances to the sediment are proportional to the sedimentation rate. I think that the text on p. 9, lines 15-17, indicates that this was done for the sensitivity analysis shown in Fig. 9. If so, it would mean that conclusions based on the analysis in Fig. 9 do not depend on the validity of the temporal variations in inputs that were fitted to the profiles. I suggest that the authors clarify this.

Reply: During the sensitivity analyses described on P.9, lines 15-17, the input of Fe oxides, Mn oxides and organic matter were changed with the same factor as the sedimentation rate. However, the temporal variations as presented in figure 3 were still applied. A doubling of the sedimentation rate would then result in a doubling of the changes presented in figure 3. For clarity we changed the text on P. 9, lines 15-17: "Subsequently, a run was performed in which the temporal input of organic matter and Fe oxides (Fig. 3) both were changed by the same factor as the sedimentation rate to account for the role of rivers as the main source of material in the region (Algesten et al. 2006, Bjorkvald et al. 2008).".

Comment #26: The effect of salinity: Section 5.4, p. 13 lines 25-27: This study highlights the role of bottom water salinity. "Because many studies focus on the relatively high salinity parts of estuaries... the role of vivianite as a sink for P has been largely overlooked." This is true. However, Hartzell et al. (2017, cited on p.2 line 35) report SEDEX P fractions along estuarine salinity gradients from 0-11 with sedimentation rates ranging from 0.4-1.8 cm yr-1. Their results were consistent with the model predictions in this paper. Although they did not identify vivianite in their sediments, they proposed that vivianite played an important role in P burial at low salinities in sediments rich in Fe2+.

Reply: We agree. Vivianite formation is indeed enhanced in environments where sulfide formation is relatively low compared to the input of Fe to the sediment. This is why we specifically included the Hartzell et al. reference in the introduction as key context for this work. We have now also included a reference to the Hartzell paper in the discussion (P.12, line 17): "as vivianite (Rozan et al., 2002; Gächter and MuÌĹller, 2003; Hartzell et al., 2017). "

Detailed comments:

Comment #27 Abstract, lines 8 and 9: Add data on the measured salinity (5) and sedimentation rates (0.25-1.0 cm yr-1) in parentheses.

Reply: The parameters, which are described at lines 8 and 9 are not based on measurements but on the modelled data, which cover a wider range than the measured in-situ data. For clarity, we prefer to not include these numbers here.

Comment #28 p. 5 line 6. Clarify what was analyzed by ICP-OES: "...dissolved... ...overnight. [The resulting solution was analyzed for] total element concentrations of..."

Reply: Sentence changed to: "Total elemental concentrations in the HNO3 solution of Al, Fe, Mn, P and S were determined by ICP-OES."

Comment #29: p. 9 line 24: "Dissolved Fe2+ and Mn2+ profiles generally showed a maximum near the sediment-water interface." This is in error. Actually, their concentrations were at a minimum near the surface, resumably due to oxidation.

Reply: The first datapoint is the bottom water sample, hence our statement here is correct.

Comment #30: p. 13, line 2: Should cite Fig. 5. Also, maxima at 42 cm are not distinct
contrary to the statement in the text.

Reply: We now refer to figure 5 on P. 13, line 2. We removed "distinct" from this sentence.

Comment #31: Fig. S. 5. Profiles of solids: Clarify the caption. Explain what FeOx 1 and FeOx 2 are. Also, note that the caption incorrectly states that the sum of these is graphed.

Reply: We have changed the caption of figure S.5 to: "Solid phase depth profiles of the easily reducible Fe oxides (Feox1) and reducible (crystalline) Fe oxides (Feox2) including vivianite, Fe carbonates (Fecarb), magnetite (Femag), elemental S (S0) and S in pyrite (CRS) for site NB8 in April 2015."

Comment #32: Consider moving Figs S.5 and S.7 to main text: They seem to present important information (S.5) and important model results (S.7). Fig S.5 seems to belong with Fig. 6. C7 Fig. S.7. seems to belong with Fig. 10 (which actually cites Fig. S.7. in the caption). Reply: Figure S.5 contains less essential extraction data. For the readability of the paper we would prefer to keep this figure in the supplements. We now combined figure S.7 and Fig. 10 to make a new Fig. 10.

New References

Dale, Andrew W., et al. "Modeling benthic–pelagic nutrient exchange processes and porewater distributions in a seasonally hypoxic sediment: evidence for massive phosphate release by Beggiatoa?." Biogeosciences 10.2 (2013): 629-651.

Jilbert, T., Asmala, E., Schröder, C., Tiihonen, R., Myllykangas, J. P., Virtasalo, J. J., ... & Hietanen, S. (2018). Impacts of flocculation on the distribution and diagenesis of iron in boreal estuarine sediments. Biogeosciences, 15, 1243-1271

Malmgren, Louise, and Lars Brydsten. "Sedimentation of river-transported particles in the Öre estuary, northern Sweden." Hydrobiologia 235.1 (1992): 59-69.

Reed, Daniel C., Caroline P. Slomp, and Gert J. de Lange. "A quantitative reconstruction of organic matter and nutrient diagenesis in Mediterranean Sea sediments over the Holocene." Geochimica et Cosmochimica Acta 75.19 (2011): 5540-5558.

Van Cappellen, Philippe, and Yifeng Wang. "Cycling of iron and manganese in surface sediments; a general theory for the coupled transport and reaction of carbon, oxygen, nitrogen, sulfur, iron, and manganese." American Journal of Science296.3 (1996): 197-243.

---

## Author Comment (AC2) · 5 Oct 2018

Review of Lenstra et al. 2018 - Variations in river input of iron impact sedimentary phosphorus burial in an oligotrophic Baltic Sea estuary Decision I would be happy for this manuscript to be published after some minor corrections.

Manuscript Quality The paper contributes knowledge of phosphorus burial a research largely overlooked in favour of carbon and nitrogen. The research focus on the main drivers of P burial in the northern Baltic sea, though a regional study this work will be of interest to a wide audience and highly relevant to Biogeosciences. The authors report that in this area a significant portion of the P burial is associated with vivianite crystalli-

sation this coupled with they're model outputs highlight an important mechanism for P. Burial where increase in fresh riverine water and iron increase P burial.

The hydrological context of this research does need better clarification.

Visual Quality Both the figures and tables are of high quality and are ready for publication.

Technical Quality The methodologies they authors used were appropriate and applied correctly, I cannot comment on modelling. I would have liked to see the core chronologies in the main text not the Sup Mat but Table.1 does provide enough information.

Reply: We thank the reviewer for the positive remarks. Below, we provide a point-by-point reply to the comments and we will revise the manuscript accordingly. The comments of reviewer #2 partly overlap with those of reviewer #1. We have indicated when and where this is the case. Comment #1: Major Comments Clarify if the flow measurement were made for the Ore river as the title and abstract suggest that the authors are directly linking river input and P burial but this is not supported by the data.

Reply: Please also see our reply to comment #1 of reviewer #1. We agree with the reviewer that measurements of riverine Fe input would form a useful addition. Unfortunately, long term measurements of total Fe input are not available for the Öre River (or any other rivers in the region) and therefore cannot be included in this manuscript. However, by modelling key porewater and solid phase depth profiles in the Öre Estuary, we demonstrate that large temporal changes in Fe and organic matter input are necessary to fit our model to the measured data at this site (P. 8-9 section 3.4). Because the study area is such an oligotrophic coastal region, the large temporal changes in Fe and organic matter input to sediments located at a river mouth can only be explained by variations in river input.

Comment #2: The data from the Ore river does differ from the averaged flows from the 86 other river is there a reason for this. Is the Ore and its catchment an oddity or is it

comparable to other estuaries in the area.

Reply: Please also see our reply to comment #3 of reviewer #1. Spatial and temporal patterns of rainfall over Sweden are expected to differ and result in differences in river discharge between the 86 rivers and the Öre River. Meltwater, for example, plays a more important role for rivers at high latitudes when compared to lower latitudes. As described on P.13 in lines 4-11, the 1996 dry period affected the entire Baltic Sea region. This was less so for 1976.

Comment #3: The authors have cores form 5 sites (NB1,6,7,8 and 10) but only model site NB8. Is there a reason for this and how comparable are the different sites. From Fig.5 it is clear that all the data falls within the same ranges but NB8 is the furthest from the river mouth and a clear statement on why the model was applied to only this site would be useful.

Reply: Please also see our reply to comment #7 of reviewer #1. In this study it was our aim to assess the burial of P in and below the SMTZ and the factors contributing to temporal variations in P burial, including the role of the rate of sedimentation. As we were not able to sample below the SMTZ at sites NB1, N6 and N10 (April) and the geochemistry of sites N7 and NB8 is comparable, we focused on site NB8. At site NB8 we subsequently performed three types of chemical extractions and experimental work to allow for visual observations of vivianite. We did not obtain these data for the other sites. The geochemistry at site NB8 is strikingly similar to that at another site with a high sedimentation rate in the Bothnian Sea (Egger et al., 2015a, b), as discussed in the manuscript (e.g. P2, lines 15-17).

We have changed the text in the model description section to clarify that we selected NB8 because it has a shallow SMTZ and a relatively high sedimentation rate (P.7 lines 3-4): "Site NB8 was characterized by a SMTZ close to the sediment-water interface and a relatively high sedimentation rate (Table 1)."

Comment #4: One important question that seems to have not been mentioned is the

potential for the Fe-P to be bound to FeOx. I think an additional figure illustrating downcore profile of Total Fe, FeOC, Fe-P and possibly vivianite bound P and organics.

Reply: In the extraction methods that we used there is not a specific step, which only extracts vivianite. In the SEDEX vivianite is extracted together with P bound to Fe oxides (Dijkstra et al., 2014, Nembrini et al., 1983). In the Fe extraction vivianite is extracted in both the HCl-Acetate and CDB step (Dijkstra et al. 2014). Therefore a single figure containing measured data of P bound to vivianite and P bound to Fe oxides is not possible. However, with the model we can make the distinction between these phases. Therefore we added a figure to the supplements (Fig. S.7), which shows the change in Fe bound in vivianite and Fe oxides as well as P bound to vivianite and Fe oxides. We now describe the figure in the results (P.10, line 29): "At depth, P bound to vivianite becomes more important compared to P bound to Fe oxides (Fig. S.7)." The relative contribution of Fe-oxide bound P and vivianite is discussed in lines 9-11 on page 12: "Due to dissolution of Fe oxides and vivianite formation in the sediment the fraction of P bound to vivianite increases with depth (Fig. S.7)."

Comment #5: Again line 4 Pg 13 – the refer to the years of 1977 and 1997 as high flow years but both looking at the Ore flow rates and S.Fig8 I would say that they have larger flows. I would focus more of the low flow years proceeding as the major mechanism.

Reply: Please also see our reply to comment #2 of reviewer #1. We agree that the river flow in the years after each relatively dry period (1976 and 1996) was not exceptionally high. The mechanisms that we propose in this manuscript as a reason for the enhanced input of Fe depend on the relatively low river flow in the years 1976 and 1996 being followed by a higher, more average, flow when directly compared to the preceding dry period. For clarity we changed the text: P2. line 25 from "In ensuing wet periods" to "After dry periods" P.13 line 5 from "In the following wet years," to "in the following years". P14 line 14 from "in a wet period directly following a dry period on land in 1976 and 1996," to "in a period following a dry year on land in 1976 and 1996."

Comment #6: Along the same lines do you have any rainfall data for this period this could be useful in further contextualising the low flows. A quick look at the UEA North Atlantic Oscillation records https://crudata.uea.ac.uk/cru/data/nao/nao.dat) both 1976 and 1996 were in the negative phase meaning dry conditions for the higher latitudes. In particular 1996 was in a very strong negative phase (NAO index:-3.27) explaining the low flows. Work completed in Scottish fjords (restricted marine environments not too dissimilar to the research area) showed that during negative NAO phases material builds up in the catchment and when the NAO switches that store of material is quickly transported to the sea – This mechanism may explain the increases in FeOx after the dry/low flow periods.

Gillibrand, PA, Cage, AG & Austin, WEN 2005, 'A preliminary investigation of basin water response to climate forcing in a Scottish fjord: evaluating the influence of the NAO' Continental Shelf Research, vol. 25, pp. 571-587. DOI: 10.1016/j.csr.2004.10.011

Reply: We agree with the reviewer that the NOA could play an important role in regulating the amount of precipitation on land, the related changes in Fe chemistry and the subsequent transfer of Fe to estuaries. We have now added a reference to the work of Gillibrand et al. (2005) and state that periods of low rainfall may be related to a negative phase in the North Atlantic Oscillation. We could not find any mention of particulate matter in the cited study, however.

Revised text (P.13 line 5): "We suggest that these high inputs of Fe and organic matter are directly linked to variations in river discharge, that may be linked to negative phases in the North Atlantic Oscillation (Gillibrand et al. 2005)".

Comment #7: The authors do focus on the role of salinity as a key component of the P burial process but as the modelling only takes place at the most saline site is the importance of this overestimated. Clarification would be useful.

Reply: The salinity at all sites was comparable ($\sim$5), therefore an overestimation is not expected. It would be interesting to also sample sites with fine-grained sediments and

a lower bottom water salinity. These types of sites were, however, not found in the Öre Estuary. To obtain insight in the role of salinity, we performed a sensitivity analysis (Fig. 9). Our results highlight that a lower salinity would increase the role of vivianite formation.

———————————————————————